# Epigenetic silencing of a multifunctional plant stress regulator

Mark Zander[1,2,3], Björn C Willige[1], Yupeng He[2], Thu A Nguyen[1†],
Amber E Langford[1], Ramlah Nehring[1‡], Elizabeth Howell[1], Robert McGrath[4§],
Anna Bartlett[2], Rosa Castanon[2], Joseph R Nery[2], Huaming Chen[2], Zhuzhu Zhang[2],
Florian Jupe[2#], Anna Stepanova[1¶], Robert J Schmitz[1**], Mathew G Lewsey[1††‡‡],
Joanne Chory[1,3], Joseph R Ecker[1,2,3]*

[1]Plant Biology Laboratory, Salk Institute for Biological Studies, La Jolla, United States; [2]Genomic Analysis Laboratory, Salk Institute for Biological Studies, La Jolla, United States; [3]Howard Hughes Medical Institute, Salk Institute for Biological Studies, La Jolla, United States; [4]Department of Biology, University of Pennsylvania, Philadelphia, United States

*For correspondence:
ecker@salk.edu

Present address: †Illumina, San Diego, United States;
‡NAVICAN, San Diego, States; §Technology Commercialization, Drexel University, Philadelphia, United States; #Bayer Crop Science, Chesterfield, United States;
¶Department of Plant & Microbial Biology, Program in Genetics, North Carolina State University, Raleigh, United States; **Department of Genetics, University of Georgia, Athens, United States;
††Department of Animal, Plant and Soil Sciences, AgriBio Building, School of Life Sciences, La Trobe University, Melbourne, Australia; ‡‡Australian Research Council Research Hub for Medicinal Agriculture, AgriBio Building, School of Life Sciences, La Trobe University, Melbourne, Australia

**Abstract** The central regulator of the ethylene (ET) signaling pathway, which controls a plethora of developmental programs and responses to environmental cues in plants, is ETHYLENE-INSENSITIVE2 (EIN2). Here we identify a chromatin-dependent regulatory mechanism at *EIN2* requiring two genes: ETHYLENE-INSENSITIVE6 (EIN6), which is a H3K27me3 demethylase also known as *RELATIVE OF EARLY FLOWERING6* (*REF6*), and EIN6 ENHANCER (EEN), the *Arabidopsis* homolog of the yeast INO80 chromatin remodeling complex subunit *IES6* (*INO EIGHTY SUBUNIT*). Strikingly, EIN6 (REF6) and the INO80 complex redundantly control the level and the localization of the repressive histone modification H3K27me3 and the histone variant H2A. Z at the 5' untranslated region (5'UTR) intron of *EIN2*. Concomitant loss of EIN6 (REF6) and the INO80 complex shifts the chromatin landscape at *EIN2* to a repressive state causing a dramatic reduction of *EIN2* expression. These results uncover a unique type of chromatin regulation which safeguards the expression of an essential multifunctional plant stress regulator.
DOI: https://doi.org/10.7554/eLife.47835.001

## Introduction

The gaseous plant hormone ethylene regulates numerous biological processes ranging from development to responses to environmental stimuli (*Johnson and Ecker, 1998*). The ET signaling pathway evolved over 450 million years ago with EIN2 as its master regulator (*Alonso et al., 1999*; *Ju et al., 2015*). In the absence of ethylene, EIN2 is constitutively phosphorylated by the Raf-like protein kinase CONSTITUTIVE TRIPLE RESPONSE1 (CTR1). Upon ET perception, EIN2 is no longer phosphorylated by CTR1, resulting in the proteolytic cleavage of EIN2's C-terminal CEND domain (EIN2C) (*Ju et al., 2012*; *Qiao et al., 2012*; *Wen et al., 2012*). This multifunctional EIN2C has multiple modes of action (*Zheng and Zhu, 2016*). In the cytoplasm, EIN2C functions to facilitate the translational repression of mRNAs for the EIN3/EIL1-degrading F-box proteins EIN3 BINDING F BOX1/2 (EBF1/EBF2) in cytoplasmic P-bodies (*Merchante et al., 2015*; *Li et al., 2015*). EIN2C also translocates to the nucleus where it stabilizes the ethylene response master regulatory transcription factors (TFs) EIN3 and EIL1 through a currently unknown mechanism (*Ju et al., 2012*; *Qiao et al., 2012*; *Wen et al., 2012*). In addition, nuclear-localized EIN2C also directly regulates histone acetylation through an association with EIN2 NUCLEAR-ASSOCIATED PROTEIN1 (ENAP1) at ET-responsive genes (*Zhang et al., 2017*). ET is crucial for the integration of biotic and abiotic stress responses into plant growth pathways (*Dubois et al., 2018*) which is underscored by the severe stress-related

phenotypes displayed by *ein2* loss-of-function alleles in *Arabidopsis* (*Alonso et al., 1999*) as well as in many other species including the legume *Medicago truncatula* and rice (*Oryza sativa*) (*Penmetsa et al., 2008*; *Ma et al., 2013*). Yet despite EIN2's importance for fitness under adverse growth conditions across land plants, its transcriptional regulation and underlying chromatin architecture are completely uncharacterized.

Here we combined genomics, genetics and biochemistry to explore the chromatin environment at *EIN2*. Through the molecular characterization of the ET-insensitive *ein6-1* mutant, we discovered that the major *Arabidopsis* H3K27me3 demethylase *RELATIVE OF EARLY FLOWERING6* (*REF6*) (*Lu et al., 2011*) and the INO80 chromatin remodeling complex cooperatively regulate H3K27me3 and H2A.Z occupancy in the 5'UTR intron of *EIN2*. Importantly, we also propose that in addition to their prominent functions (H3K27me3 demethylation and H2A.Z eviction), EIN6 (REF6) and the INO80 complex also antagonize SWR1-mediated H2A.Z incorporation and PRC2-mediated tri-methylation of H3K27, respectively. Altogether, our results uncover an epigenetic control mechanism that establishes a permissive chromatin state at a critical plant stress regulator.

## Results

### Mutations in two different genes are responsible for the ET-insensitivity of the *ein6-1* mutant

The vast majority of key ET signaling components have been discovered through genetic screens exploiting the triple response phenotype of ET-treated dark-grown seedlings (*Guzmán and Ecker, 1990*). The *ein6-1* mutant originates from a genetic screen of fast neutron-mutagenized *A. thaliana* Landsberg *erecta* (L*er*) seeds (*Roman et al., 1995*). However, the causal mutation in the original, ET-insensitive, *ein6-1* mutant has remained elusive for more than two decades. Surprisingly, SHOREmap (*Schneeberger et al., 2009*) and segregation analyses identified mutations in two different genes in the *ein6-1* mutant (*Figure 1A*; *Figure 1—figure supplement 1A*). Interestingly, only one of the two single mutants displays an ET-hyposensitive root phenotype which we refer to as the *ein6-1* single mutant (*Figure 1A*). The second gene mutation was named *EIN6 ENHANCER* (*EEN*) because while *een-1* plants show no phenotype, *ein6-1 een-1* double mutants show an ET-insensitive root phenotype (*Figure 1A*).

DNA sequencing revealed that the causal mutation at the *ein6-1* allele is a seven base pair (bp) deletion in the fifth exon of the *RELATIVE OF EARLY FLOWERING6* (*REF6, At3g48430*) gene, leading to a premature stop codon (*Figure 1—figure supplement 1B*). *REF6* encodes the major H3K27me3 demethylase in *Arabidopsis* (*Lu et al., 2011*). The *een-1* mutation was discovered by Thermal Asymmetric Interlaced (TAIL)-PCR and also directly visualized using optical mapping of ultra-long L*er* and *ein6-1 een-1* DNA molecules (*Figure 1B*; *Figure 1—figure supplement 1C and D*). *een-1* arose from a genomic inversion event of a 83 kilobase pair (kb) region on chromosome four. As a result of two fast neutron bombardment-induced double-strand breaks, one in the 5'UTR of *EEN* (*At4g38495*) and the other in the fourth intron of the *AQUAPORIN INTERACTOR* (*AQI, At4g38220*) gene, the entire *EEN* gene was fused with four exons of the *AQI* gene (*Figure 1—figure supplement 1C–G*). Sequencing of the chimeric *AQI-EEN* genomic DNA and cDNA revealed that 14 bp of the *EEN* 5'UTR are now part of the *AQI-EEN* cDNA causing an open reading frame shift that results in a premature stop codon within the first exon of *EEN* (*Figure 1—figure supplement 1E–G*). Interestingly, *EEN* shares homology to yeast *IES6*, a crucial component of the yeast INO80 chromatin remodeling complex whose function is linked to eviction of the histone variant H2A.Z (*Papamichos-Chronakis et al., 2011*; *Chen et al., 2013*) (*Figure 2—figure supplement 1A*).

To explore the interplay between EIN6 (REF6) and EEN in regulating the ET pathway, we examined the transcriptional response to ET using RNA-seq (*Supplementary file 1*). While both single mutants show nearly wild-type transcriptional responses to ET, *ein6-1 een-1* double mutants are similar to *ein2-45* mutants, being largely deficient in activating ET-dependent transcription (*Figure 1C*). A hallmark of an activated ET signaling pathway is EIN2C-mediated stabilization of the transcription factors EIN3 and EIL1 (*Guo and Ecker, 2003*; *Potuschak et al., 2003*; *Qiao et al., 2012*; *Wen et al., 2012*; *Ju et al., 2012*). To confirm a previous report showing that the levels of stabilized EIN3 are drastically reduced in *ein6-1 een-1* mutants (*Guo and Ecker, 2003*), immunoblots using an anti-EIN3 antibody were conducted. While wild-type and single mutants accumulate similar levels of EIN3,

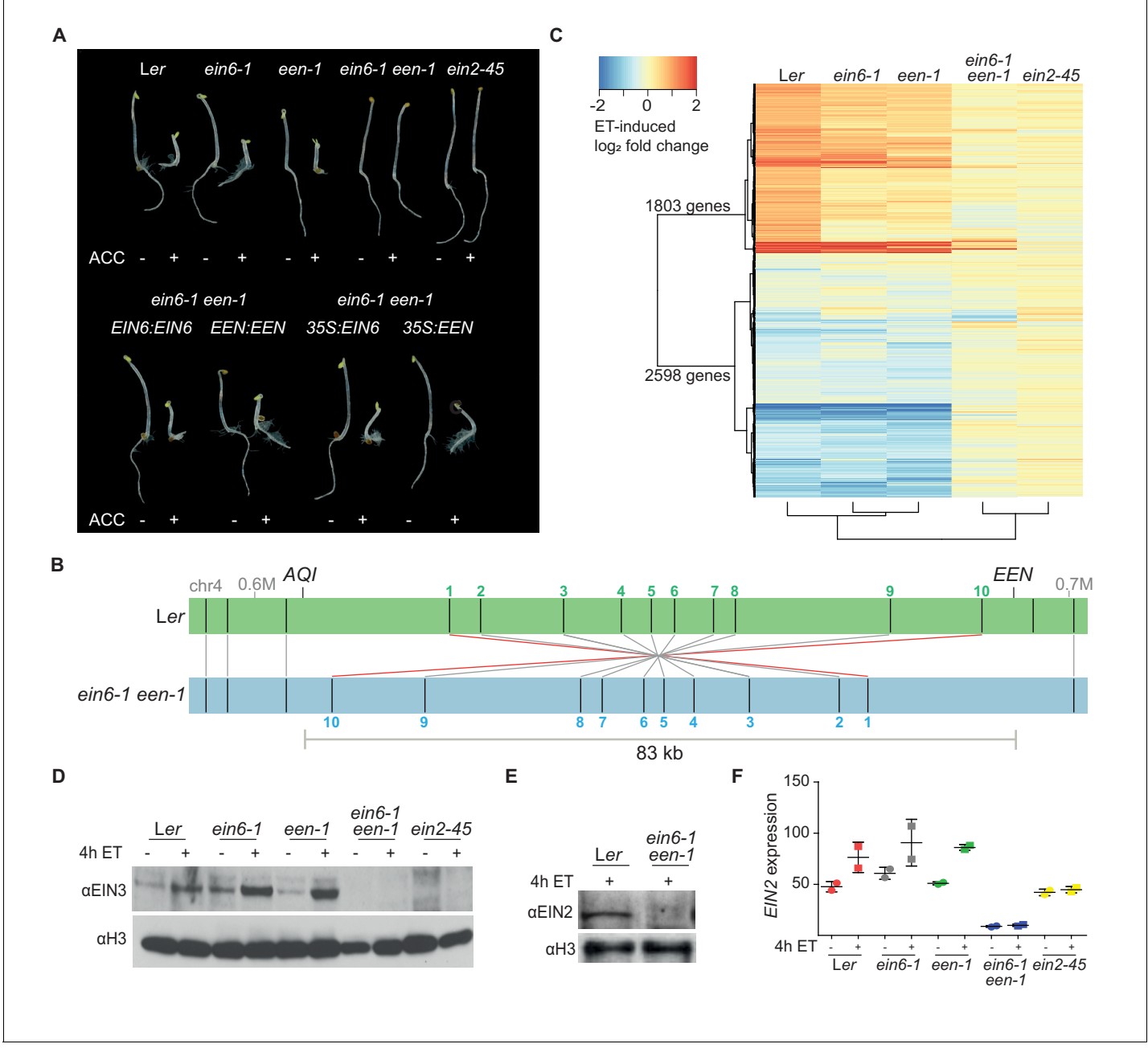

**Figure 1.** Mutations in two different genes are responsible for the ET-insensitivity in *ein6-1* plants. (**A**) Triple response phenotype of 3-day-old etiolated seedlings of L*er*, *ein6-1*, *een-1*, *ein6-1 een-1*, *ein2-45* (upper row) and of the indicated complementation lines, either driven by the respective native promoter or by the Cauliflower mosaic virus *35S* promoter (*35S*) (lower row). Seedlings were grown on control LS media or LS media supplemented with 10 μM ACC. (**B**) Schematic illustration of a Bionano Genomics Irys optical map of the *een-1* inversion region at the end of the fourth chromosome in *ein6-1 een-1* (blue) aligned to an optical map of L*er* (green). Original output is shown in *Figure 1—figure supplement 1D*. Nick sites are indicated as black lines within the respective optical map. Matching nick sites between the maps are indicated as gray lines. Numbered nick sites are used to better visualize the inversion event. The approximate position of *EEN* and *AQI* is indicated as well. (**C**) Heatmap visualizes the log₂ fold change of expression in L*er*, *ein6-1*, *een-1*, *ein6-1 een-1* and *ein2-45* seedlings in response to 4 hr of ethylene (ET) treatment detected by RNA-seq. Differentially expressed genes (DE genes) that are significantly induced (1803 genes) or repressed (2598 genes) after 4 hr of ET treatment in L*er* seedlings are shown. Cluster dendogram below the heatmap indicate similarities between the tested genotypes. (**D**), (**E**), Western blot analysis of nuclear extracts of 3-day-old etiolated L*er*, *ein6-1*, *een-1*, *ein6-1 een-1* and *ein2-45* seedlings that were either treated for 4 hr with hydrocarbon-free air (control) or ET gas. Antibodies against EIN3 and EIN2 were used to detect nuclear-localized EIN3 (**D**) and EIN2C (**E**), respectively. Amounts of histone H3 were detected with an anti-H3 antibody and served as a loading control. (**F**) *EIN2* expression in response to 4 hr of ET treatment in the indicated genotypes using

*Figure 1 continued on next page*

*Figure 1 continued*

RNA-seq is shown. *EIN2* transcripts can still be detected in *ein2-45* mutants because they only harbor a point mutation in the C-terminus of EIN2 (*Beaudoin et al., 2000*). Expression is measured in TPM (Transcripts Per Kilobase Million) and results from two biological replicates are shown.

DOI: https://doi.org/10.7554/eLife.47835.002

The following figure supplement is available for figure 1:

**Figure supplement 1.** Characterization of the two causal mutations in *ein6-1 een-1* double mutants.

DOI: https://doi.org/10.7554/eLife.47835.003

*ein6-1 een-1* mutants accumulate no EIN3 protein in response to ET treatment (*Figure 1D*). Another prerequisite of the ET response is nuclear localization of the EIN2C domain (*Qiao et al., 2012*; *Wen et al., 2012*; *Ju et al., 2012*). However, nuclear-localized EIN2C protein is absent in *ein6-1 een-1* mutants suggesting that the failure to accumulate EIN3 is due to the absence of EIN2C (*Figure 1E*). Results from RNA-seq indicate that this absence can be explained by the reduction of *EIN2* mRNA expression in *ein6-1 een-1* double mutants (*Figure 1F*).

## A repressive chromatin environment at *EIN2* down-regulates its expression

Based on the known H3K27me3 demethylase function of EIN6 (REF6) (*Lu et al., 2011*) and the potential role of EEN in chromatin remodeling, we hypothesized that the observed impaired *EIN2* expression in *ein6-1 een-1* mutants is manifested through a specific chromatin-related process. This hypothesis was tested by employing chromatin immunoprecipitation sequencing (ChIP-seq) and MethylC-seq to characterize histone modifications and cytosine DNA methylation, respectively (*Supplementary file 2*). We first analyzed the genome-wide occupancy of H3K27me3 and discovered a genome-wide ectopic gain of H3K27me3 in the *ein6-1* mutant which was previously reported for *ref6-3* mutants (*Lu et al., 2011*). There were 2369 genes (Group I) that gain H3K27me3 in *ein6-1* single mutants (2-fold enrichment over L*er*) as well as in *ein6-1 een-1* double mutants (*Figure 2—figure supplement 1B and C*; *Supplementary file 3*). We also found 54 additional genes (Group II) with significantly higher H3K27me3 levels (2-fold enrichment over *ein6-1*) in the double mutant (*Figure 2—figure supplement 1C and D*; *Supplementary file 3*). Strikingly, *EIN2* is in Group II and displays one of the strongest increases of H3K27me3 in *ein6-1 een-1* mutants, with the mark spreading throughout the gene body (*Figure 2A*; *Figure 2—figure supplement 1E*; *Supplementary file 3*).

Given the role of the INO80 complex in H2A.Z eviction (*Alatwi and Downs, 2015*; *Papamichos-Chronakis et al., 2011*), we employed ChIP-seq to survey the genome-wide distribution of H2A.Z using a commercial H2A.Z antibody (*Jupe et al., 2019*). The Swi2/Snf2-related 1 (SWR1) complex subunit PHOTOPERIOD-INDEPENDENT EARLY FLOWERING1 (PIE1) is required for H2A.Z deposition (*Noh and Amasino, 2003*) and ChIP-seq of H2A.Z in *pie1-1* mutants confirmed a lack of H2A.Z genome-wide (*Figure 2—figure supplement 1G and H*). To further validate our H2A.Z antibody, we performed ChIP-seq analyses with equal amounts of the same Col-0 *HTA11:HTA11-GFP* chromatin using either a commercial GFP or our commercial H2A.Z antibody. Both antibodies gave similar H2A.Z occupancy results (*Figure 2—figure supplement 1F–H*). In addition, a direct comparison of our antibody validation H2A.Z profiles (Col-0, *pie1-1*, Col-0 *HTA11:HTA11-GFP* (αH2A.Z) and Col-0 *HTA11:HTA11-GFP* (αGFP)) with three publicly available H2A.Z ChIP-seq datasets (*Carter et al., 2018*; *Wollmann et al., 2017*; *Coleman-Derr and Zilberman, 2012*) further confirmed the suitability of our H2A.Z antibody (*Figure 2—figure supplement 1F–H*). Only 17 genes showed an increase of H2A.Z in *ein6-1 een-1* mutants (1.5-fold enrichment over *een-1*), and again, *EIN2* was among genes having high levels of ectopically localized H2A.Z throughout its entire gene body (*Figure 2A*, *Figure 2—figure supplement 1I*; *Supplementary file 4*). H2A.Z and H3K27me3 are functionally linked and both repress transcription in *Arabidopsis* (*Carter et al., 2018*; *Coleman-Derr and Zilberman, 2012*). These findings imply that the strong accumulation of H3K27me3 and H2A.Z at *EIN2* creates a highly repressive chromatin environment leading to strongly reduced *EIN2* expression and ET-insensitivity in *ein6-1 een-1* mutants (*Figures 1F* and *2A*).

Concordant with its compromised expression, levels of H3K4me3, a histone mark that is typically enriched at transcriptionally active genes in *Arabidopsis* (*Fromm and Avramova, 2014*), were reduced at *EIN2* in the double mutant (*Figures 1F* and *2A*; *Figure 2—figure supplement 2A*;

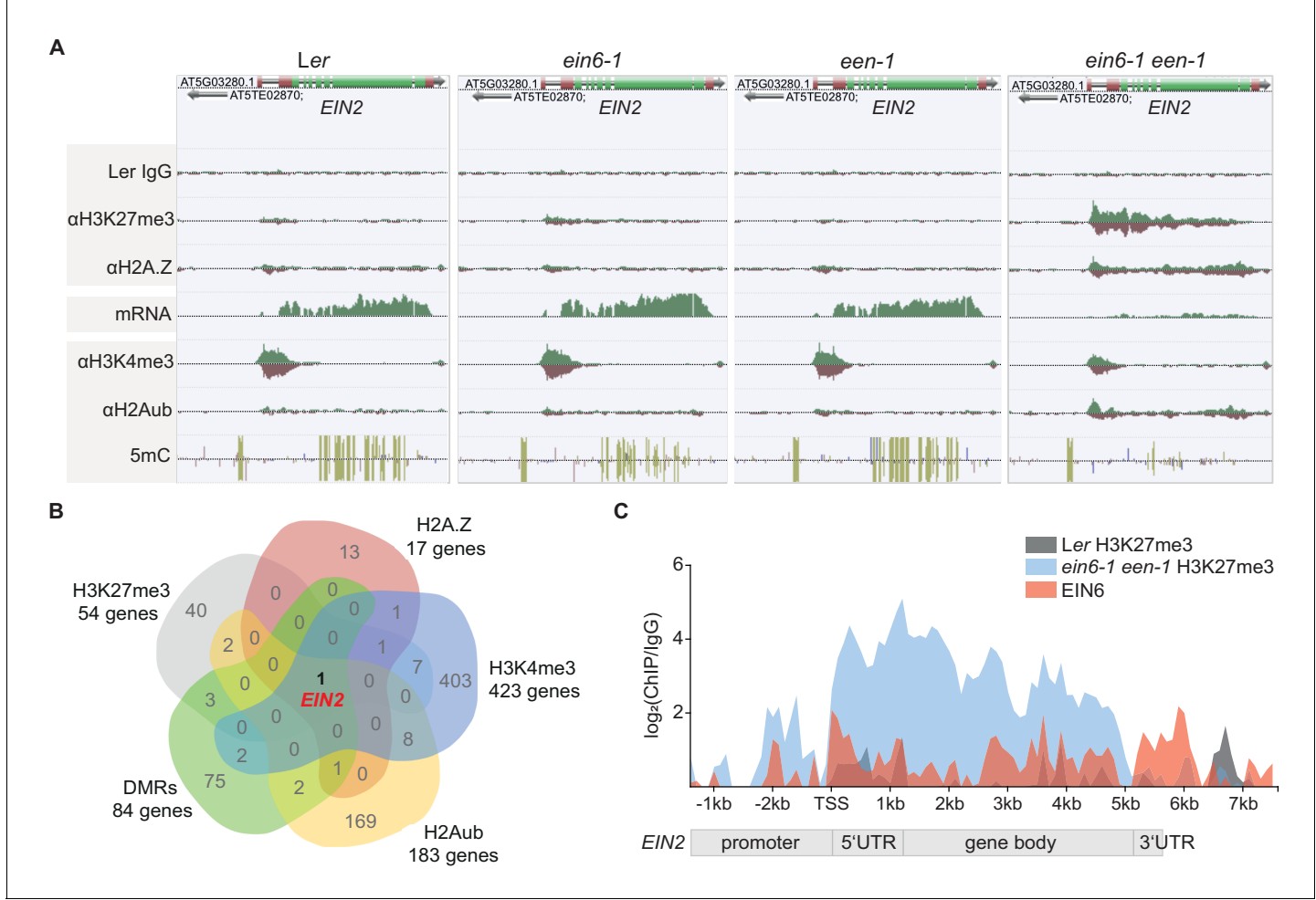

**Figure 2.** A repressive chromatin environment at *EIN2* down-regulates its expression. (**A**) Genome browser screenshot visualizes the levels of the depicted chromatin features at the *EIN2* gene in untreated 3-day-old etiolated L*er*, *ein6-1*, *een-1* and *ein6-1 een-1* seedlings. Occupancy of H3K27me3, H2A.Z, H3K4me3 and H2Aub was determined with ChIP-seq, mRNA expression was measured with RNA-seq and levels of methylated cytosines (CG in yellow, CHG in blue, CGG in pink) were determined with MethylC-seq. To ensure an accurate comparison of individual chromatin features between genotypes, the tracks were normalized to the respective sequencing depth. Normalization was separately done for each chromatin feature. Biological replicate 1 of the H3K27me3 and H2A.Z ChIP-seq datasets is shown. (**B**) Venn diagram illustrates the overlap between genes that show a significant increase of H3K27me3 (2-fold enrichment over *ein6-1*), H2AZ (2-fold enrichment over *een-1*) and H2Aub (2-fold enrichment over L*er*) in *ein6-1 een-1* mutants and also a significant decrease of H3K4me3 (1.5-fold enrichment in L*er* over *ein6-1 een-1*). In addition, genes that contain differentially methylated regions (DMRs) with ten or more methylated cytosines in at least one genotype were included as well. (**C**) Graphical illustration of H3K27me3 and EIN6 occupancy at the *EIN2* gene determined with ChIP-seq. Sequencing reads were merged between biological replicates for the H3K27me3 ChIP-seq using untreated 3-day-old etiolated L*er* (gray) and *ein6-1 een-1* (blue) seedlings (two replicates each) and the EIN6 ChIP-seq (red) using L*er* *35S:EIN6-FLAG* seedlings (three replicates). The occupancy was calculated as the ratio between the respective merged ChIP and the merged L*er* IgG control in 100 bp bins from 2.4 kb upstream to 7.7 kb downstream of the transcriptional start site (TSS) of *EIN2* and is shown as $\log_2$ fold change. Negative values which reflect lower occupancy in the ChIP sample compared to the IgG control sample were set to zero.

DOI: https://doi.org/10.7554/eLife.47835.004

The following figure supplements are available for figure 2:

**Figure supplement 1.** Quantitative description of the chromatin environment at *EIN2* in *ein6-1 een-1* mutants.
DOI: https://doi.org/10.7554/eLife.47835.005

**Figure supplement 2.** *EIN2* displays an aberrant chromatin signature *ein6-1 een-1* mutants and is a direct target of EIN6.
DOI: https://doi.org/10.7554/eLife.47835.006

*Supplementary file 4*). In keeping with the accumulation of repressive marks, levels of mono-ubiquitinated H2A (H2Aub) are also clearly increased at *EIN2* in *ein6-1 een-1* mutants, demonstrating increased PRC1-mediated gene silencing (*Figure 2A*; *Figure 2—figure supplement 2B*; *Supplementary file 4*). Moreover, MethylC-seq analysis also discovered a complete loss of gene body cytosine DNA methylation at *EIN2* in the double mutant (*Figure 2A*; *Figure 2—figure supplement 2C*; *Supplementary file 4*); the accumulation of gene body H2A.Z is mutually exclusive with DNA methylation (*Coleman-Derr and Zilberman, 2012*). These results reveal the highly specific nature of EIN6 (REF6)/EEN-mediated regulation since *EIN2* is the only gene in the entire *Arabidopsis* genome displaying such a unique chromatin change in *ein6-1 een-1* mutants (*Figure 2B*).

EIN6 (REF6) binds to the CTCTGYTY DNA motif at target genes through its C-terminal zinc fingers (*Cui et al., 2016*; *Li et al., 2016*). The 5'UTR intron region of *EIN2* contains two of these DNA motifs (*Figure 2—figure supplement 2D*). Therefore, we tested for direct EIN6/EIN6-ZNF binding to *EIN2* by conducting EIN6 ChIP-seq and EIN6 zinc finger (EIN6-ZNF) DAP-seq (*O'Malley et al., 2016*) assays. Both assays successfully identified the CTCTGYTY motif as the preferred binding motif (*Figure 2—figure supplement 2E and F*; *Supplementary file 5*). We observed direct EIN6/EIN6-ZNF binding within the 5'UTR intron and the gene body of *EIN2* (*Figure 2C*; *Figure 2—figure supplement 2G*). The importance of the zinc fingers of EIN6 (REF6) for *EIN2* regulation was further confirmed by the failure of a truncated *EIN6* (*REF6*) construct without the zinc fingers (*EIN6ΔZNF*) to complement the *ein6-1 een-1* triple response phenotype (*Figure 2—figure supplement 2H*).

## EEN is a subunit of the *Arabidopsis* INO80 chromatin remodeling complex

Next, we focused on the molecular characterization of *EEN*, an uncharacterized *Arabidopsis* gene. EEN is a small YL1 nuclear protein C-terminal (YL1-C) domain-containing 14 kilodalton (kDa) protein that is cytoplasm- and nuclear-localized and expressed throughout the entire *Arabidopsis* seedling (*Figure 2—figure supplement 1A*; *Figure 3—figure supplement 1A and B*). EEN also binds to *EIN2*, especially in the 3'UTR region (*Figure 3A*; *Figure 3—figure supplement 1C*). Its yeast homolog IES6 forms the nucleosome recognition module of the INO80 chromatin remodeling complex with ACTIN-RELATED PROTEIN5 (ARP5) (*Eustermann et al., 2018*). This module is crucial for remodeling activity of the INO80 complex (*Tosi et al., 2013*; *Eustermann et al., 2018*; *Watanabe et al., 2015*). In vitro pulldowns demonstrated an interaction between EEN and *Arabidopsis* ARP5 (*Figure 3B*). To investigate whether this interaction also occurs in planta, mass spectrometry (MS) analysis was conducted which identified ARP5 as a strong EEN interactor (*Figure 3—figure supplement 1D*; *Supplementary file 6*). Next, we validated the INO80 complex subunit function of EEN by analyzing the transcriptomes of newly identified *ino80* and *een* T-DNA insertion mutants (*ino80-8*, *een-2*) (*Figure 3—figure supplement 1E–H*). Compared to wild-type plants, 1866 genes were differentially expressed in *ino80-8* mutants (*Figure 3C*). These same genes were also differentially expressed in a remarkably similar manner in *een-2* and *arp5-1* mutants (*Figure 3C*; *Supplementary file 7*). Finally, we generated double mutants (*ref6-1 een-2*, *ref6-1 ino80-1* and *ref6-1 arp5-1*) that each strikingly display increased levels of H3K27me3 and H2A.Z at *EIN2* (*Figure 3D*, *Figure 3—figure supplement 1I and J*) highlighting that the *ein6-1 een-1* molecular phenotype cannot be attributed to the unique combination of a fast-neutron bombardment-induced deletion (*ein6-1*) and inversion (*een-1*). These double mutants also exhibit a hypo-sensitive triple response phenotype, although not as severe as in *ein6-1 een-1* mutants (*Figure 3—figure supplement 1K*). We speculate that differences between ecotypes (Ler versus Col-0) and/or differences between the *ein6/ref6* alleles (7 bp deletion versus T-DNA insertion) might cause this phenotypic discrepancy. Interestingly, *ref6-1 een-2 pie1-1* triple mutants that are also deficient in the ATPase subunit of the SWR1 chromatin remodeling complex display a wild-type triple response phenotype (*Noh and Amasino, 2003*). This result suggests that the loss of chromatinized H2A.Z in *pie1-1* mutants can suppress the hypo-sensitive triple response phenotype of *ref6-1 een-2* mutants (*Figure 2—figure supplement 1G and H*; *Figure 3—figure supplement 1K*). Taken together, our combined biochemical, genomic and genetic data identified EEN, complexed with ARP5, an integral part of the *Arabidopsis* INO80 chromatin remodeler along with EIN6 (REF6), directly controls the chromatin environment at *EIN2*.

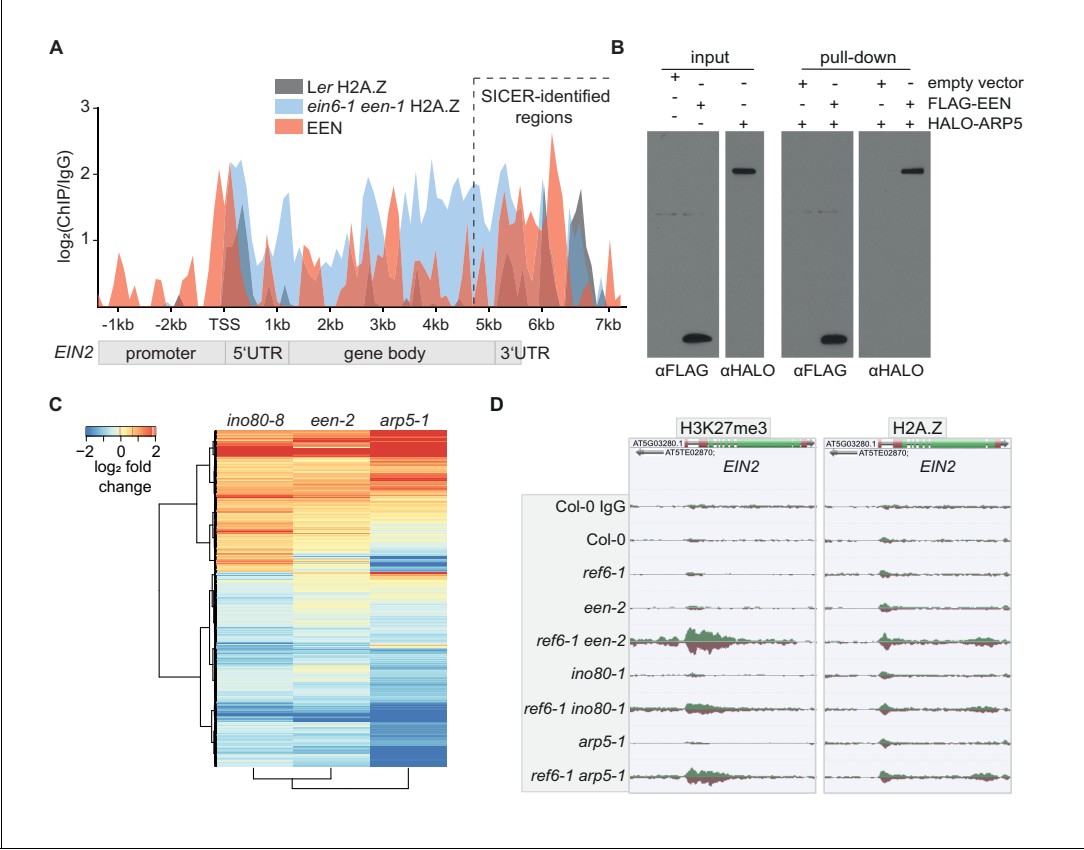

**Figure 3.** EEN is a subunit of the INO80 chromatin remodeling complex. (**A**) Graphical representation of H2A.Z and EEN occupancy at the *EIN2* gene determined with ChIP-seq. Sequencing reads of two merged H2A.Z ChIP-seq's (L*er* (gray), *ein6-1 een-1* (blue)) and one EEN ChIP-seq (Col-0 *35S:EEN-FLAG* (red)) using 3-day-old etiolated seedlings were used. The occupancy is displayed as log₂ fold change and was calculated as the ratio between the respective ChIP and the respective IgG control in 100 bp bins from 2.4 kb upstream to 7.7 kb downstream of the transcriptional start site (TSS) of *EIN2*. Occupancies that were lower in the ChIP sample compared to the IgG sample were set to zero. The region with significant EEN enrichment determined with SICER is indicated as well. (**B**) FLAG-pull-down assays using FLAG-tagged EEN and Halo-tagged ARP5 that were in vitro translated in rabbit reticulocyte extract. Anti-FLAG and anti-Halo antibodies were used to confirm EEN/ARP5 expression, EEN immunoprecipitation and EEN-ARP5 interaction. (**C**) Heatmap illustrates the log₂ fold change in mRNA expression of INO80-dependent genes in untreated 3-day-old etiolated *ino80-8*, *een-2* and *arp5-1* seedlings relative to their expression in Col-0 seedlings. INO80-dependent genes were selected as genes that show a significant differential expression in *ino80-8* mutant seedlings compared to Col-0 seedlings (787 genes up, 1079 genes down). (**D**) Genome browser screenshot shows the occupancy of H3K27me3 (right) and H2A.Z (left) at the *EIN2* gene in untreated 3-day-old etiolated Col-0, *ref6-1*, *een-2*, *ref6-1 een-2*, *ino80-1*, *ref6-1 ino80-1*, *arp5-1* and *ref6-1 arp5-1* seedlings. Genome-wide occupancy of H3K27me3 and H2A.Z was determined by ChIP-seq. To ensure an accurate comparison of individual chromatin features between genotypes, the tracks were normalized to the respective sequencing depth. Normalization was separately done for each chromatin feature.

DOI: https://doi.org/10.7554/eLife.47835.007

The following figure supplement is available for figure 3:

**Figure supplement 1.** Proteomic and genetic validation of EEN as a subunit of the *Arabidopsis* INO80 chromatin remodeling complex.
DOI: https://doi.org/10.7554/eLife.47835.008

## INO80 controls ethylene-induced H2A.Z eviction dynamics

INO80 interacts with H2A.Z in *Arabidopsis* (*Zhang et al., 2015*) and high levels of H2A.Z at *EIN2* in *ein6-1 een-1* mutants suggest a role for the *Arabidopsis* INO80 complex in controlling H2A.Z dynamics as reported for other organisms (*Figure 2A*) (*Papamichos-Chronakis et al., 2011*; *Alatwi and Downs, 2015*). Surprisingly, a global defect of the H2A.Z landscape in *ino80-1* mutants and mutants of INO80 complex subunits (*een-1*, *een-2* and *arp5-1*) to wild-type was not observed (*Figure 4A and B*). Except for *pie1-1* mutants, the global impact on the H2A.Z landscape was only marginally affected in all tested single and double mutants (*Figure 4A and B*; *Figure 2—figure supplement 1G and H*). Furthermore, the gain of H2A.Z only becomes evident in double mutants at a few loci

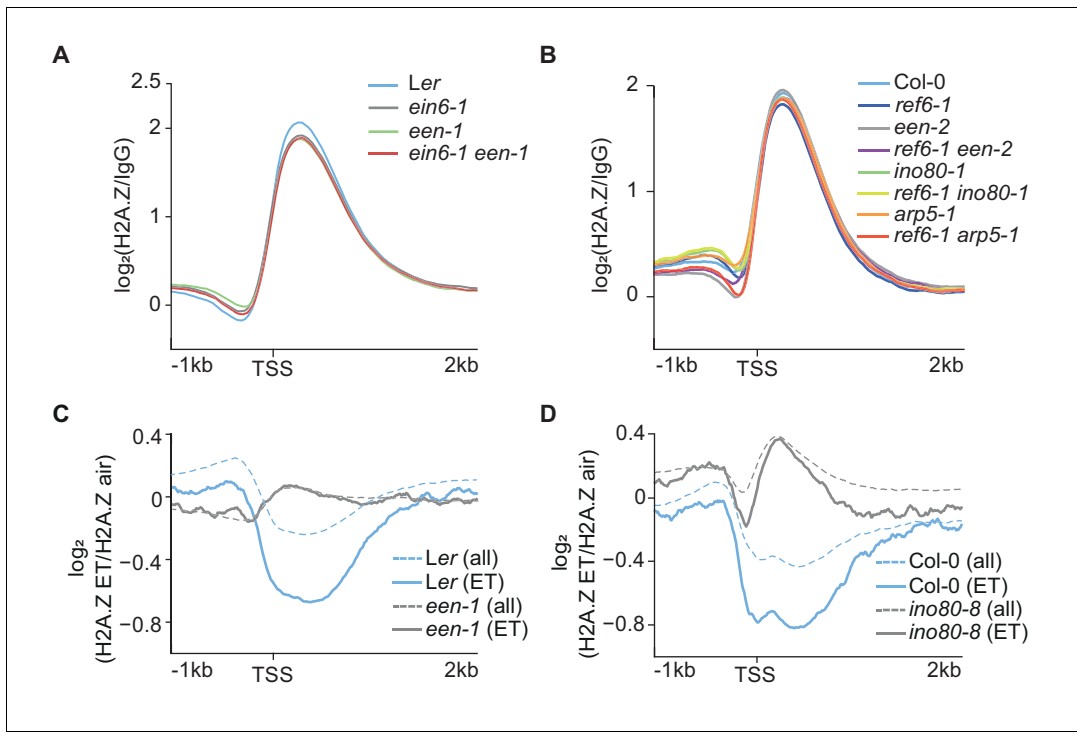

**Figure 4.** INO80 controls ethylene-induced H2A.Z eviction dynamics. (A) (B) Aggregated H2A.Z profile of all *Arabidopsis* genes (TAIR10) in L*er*, *ein6-1*, *een-1*, *ein6-1 een-1* (A) and Col-0, *ref6-1*, *een-2, ref6-1 een-2, ino80-1, ref6-1 ino80-1, arp5-1* and *ref6-1 arp5-1* (B) seedlings from 1 kb upstream to 2 kb downstream of the TSS determined by ChIP-seq. (C) (D) ET-induced H2A.Z eviction dynamics are shown as aggregated H2A.Z profiles of 3-day-old etiolated L*er*, *een-1*, Col-0 and *ino80-8* seedlings treated for 4 hr with ET gas. Two biological H2A.Z ChIP-seq replicates for L*er* and *een-1* (C) and one for Col-0 and *ino80-8* (D) were analyzed. Profiles visualize the log₂ fold change between the respective air and ET-treated H2A.Z ChIP-seq samples from 1 kb upstream to 2 kb downstream of the TSS. 1076 genes that show a robust H2A.Z eviction in response to ET in all replicates (≥1.3 fold SICER comparison, air vs 4 hr ET) are shown as solid lines whereas all genes (TAIR10) are shown as dashed lines.
DOI: https://doi.org/10.7554/eLife.47835.009

The following figure supplement is available for figure 4:

**Figure supplement 1.** EEN binds to gene bodies of environmentally responsive genes.
DOI: https://doi.org/10.7554/eLife.47835.010

including *EIN2* (*Figures 2A* and *3D*). Interestingly, ChIP-seq assays revealed that EEN predominantly binds to gene bodies of environmentally responsive genes that harbor gene body H2A.Z (*Figure 4—figure supplement 1A–C*; *Supplementary file 5*). Considering the role of H2A.Z in conferring gene responsiveness to environmental stimuli (*Sura et al., 2017*; *Coleman-Derr and Zilberman, 2012*), we speculated that the INO80 complex is potentially involved in the stimulus-induced eviction of H2A.Z. Thus, we examined the H2A.Z landscape in *een-1* and *ino80-8* mutants in ET-treated seedlings and found that robust hormone-induced H2A.Z eviction (≥1.3 fold, air vs 4 hr ET) occurs at 1076 genes (*Supplementary file 8*). These eviction dynamics were compromised in *een-1* and *ino80-8* mutants suggesting that the *Arabidopsis* INO80 complex is involved in the H2A.Z eviction process (*Figure 4C and D*; *Figure 4—figure supplement 1D*).

## EIN6 (REF6)/INO80-mediated regulation occurs at the 5'UTR intron region of *EIN2* during embryogenesis

EIN6 (REF6) can associate with INO80 in vivo (*Smaczniak et al., 2012*). To investigate when the EIN6 (REF6)/INO80-mediated regulation of *EIN2* occurs, we used ChIP-seq to map the H3K27me3 landscape in various tissues representing different developmental stages of wild-type and *ein6-1 een-1* (dry seeds, roots and shoots of etiolated seedlings, de-etiolated whole-seedlings, flowers and

siliques) (*Figure 5A*). By using the broad H3K27me3 domain at *EIN2* as a proxy for an aberrant chromatin signature in *ein6-1 een-1* mutants, we discovered its presence in all tested *ein6-1 een-1* tissues (*Figure 5A*). These results indicate that EIN6 (REF6)/INO80-mediated control of *EIN2* expression already occurs by early embryogenesis in wildtype and that its failure rapidly establishes a repressive chromatin state at *EIN2*.

A key feature of the broad H3K27me3 domain at *EIN2* in *ein6-1 een-1* mutants is its asymmetric distribution across the gene with the peak level occurring within a 5'UTR intron and a steady decrease towards the 3'UTR (*Figures 2A* and *5A*). The 5'UTR intron is also the region of *EIN2* where the majority of H3K27me3, H2A.Z, H2Aub and H3K4me3 can be detected in wild-type plants (*Figure 2A*). To interrogate the role of the EIN2 5'UTR intron in more detail, we generated *EIN2* promoter *GUS* fusions with or without the 5'UTR intron (*EIN2:GUS* and *EIN2ΔUI:GUS*, respectively) and introduced these constructs into wild-type and *ein6-1 een-1* plants. Histochemical GUS staining revealed detectable reporter activity from the *EIN2:GUS* construct in wild-type but only minimal activity was observed in *ein6-1 een-1* seedlings (*Figure 4B*). This result recapitulates the differences observed in endogenous *EIN2* mRNA expression between the two genotypes (*Figures 1F* and

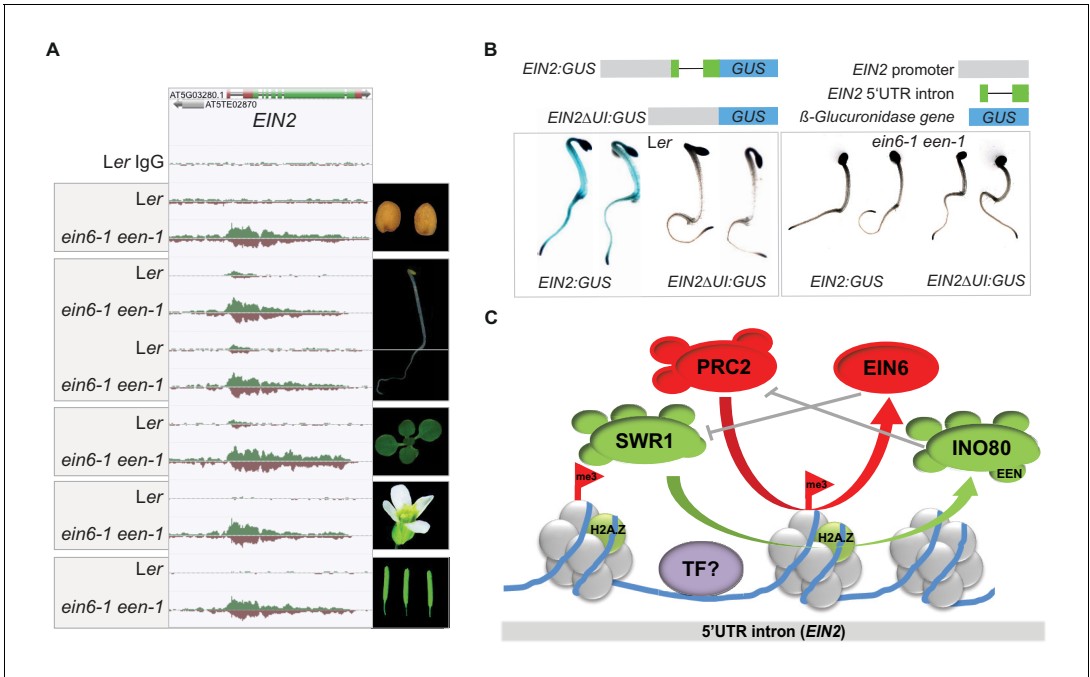

**Figure 5.** EIN6 (REF6)/INO80-mediated regulation occurs at the 5'UTR intron region of *EIN2* during embryogenesis. (A) Genome browser screenshot visualizes levels of H3K27me3 at the *EIN2* gene in the indicated tissues of L*er* and *ein6-1 een-1* plants (dry seeds, roots and shoots of etiolated seedlings, de-etiolated whole-seedlings, flowers and siliques). H3K27me3 occupancy was identified with ChIP-seq. The tracks were normalized to the respective sequencing depth for each experiment to allow an accurate comparison between L*er* and *ein6-1 een-1* H3K27me3 profiles. Normalization was separately done for each chromatin feature. (B) Histochemical GUS staining of untreated three-day-old etiolated L*er* and *ein6-1 een-1* seedlings stably expressing either a *EIN2* promoter *GUS* fusions with (*EIN2:GUS*) or without the 5'UTR intron (*EIN2ΔI:GUS*). Schematic illustration of the used *EIN2:GUS* constructs is shown as well. The gray box indicates the 2 kb promoter region of *EIN2*, the green boxes represent the 5'UTR and the black line indicates the 5'UTR intron. (C) Hypothetic model of the EIN6 (REF6)/INO80-mediated double safeguard mechanism at the 5'UTR intron of *EIN2*. Besides the demethylase function role of EIN6 (REF6) in antagonizing PRC2-mediated tri-methylation of H3K27, we speculate that EIN6 (REF6) also antagonizes SWR1-facilitated H2A.Z incorporation at *EIN2*. This scenario of a dual function holds true for the INO80 complex as well which counteracts SWR1 by removing H2A.Z, but also antagonizes H3K27 trimethylation through an unknown mechanism. The resulting low levels of H3K27me3 and H2A. Z possibly increase the accessibility of the enhancer element for binding of yet unknown transcription factors (TFs). When this EIN6 (REF6)/INO80-mediated regulation is functionally impaired, H3K27me3 and H2A.Z start to accumulate over the entire gene body of *EIN2* mutually reinforcing their depositions.

DOI: https://doi.org/10.7554/eLife.47835.011

The following figure supplement is available for figure 5:

**Figure supplement 1.** The 5'UTR intron region of *EIN2* is crucial for *EIN2* expression.
DOI: https://doi.org/10.7554/eLife.47835.012

*2A*). Deletion of the 5'UTR intron resulted in strongly reduced GUS reporter activity in wild-type and *ein6-1 een-1 EIN2ΔUI:GUS* seedlings (*Figure 5B*), indicating that, not only is the 5'UTR intron necessary for *EIN2* promoter-driven reporter activity, but also that its loss phenocopies the *ein6-1 een-1* mutant. We further investigated the 5'UTR intron using transient quantitative GUS reporter assays in *Nicotiana benthamiana* leaves driven by full-length and 3'-truncated intron 5'UTR constructs. We found that the full length *EIN2* 5'UTR intron construct ($UI^{1161}$:GUS) as well as two moderately 3'-truncated constructs ($UI^{703}$:GUS and $UI^{404}$:GUS) were able to drive GUS reporter activity to comparable levels (*Figure 5—figure supplement 1A*). However, the most heavily 3'-truncated intron construct ($UI^{170}$:GUS) had greater than six-fold higher GUS activity relative to the longer constructs, indicating the presence of an enhancer element in that region whose activity may be modulated by one or multiple regulatory elements further downstream in the 5'UTR intron region (*Figure 5—figure supplement 1A*). Similar GUS activities of $UI^{1161}$:GUS, $UI^{703}$:GUS and $UI^{404}$:GUS suggest that these regulatory elements are located within the +170 bp to +404 bp region and provide a landing platform for various, but yet unknown, transcriptional regulators. By interrogating the Plant Cistrome Database (*O'Malley et al., 2016*), we identified bHLH122 and SQUAMOSA PROMOTER BINDING PROTEIN-LIKE 5 (SPL5) as potential candidate transcription factors (*Figure 5—figure supplement 1B*). It is known from SPL15 that it associates with the MADS-box protein SOC1 which in turn recruits EIN6 (REF6) to remodel the chromatin environment at SPL15/SOC1 target genes (*Hyun et al., 2016*).

## Discussion

Here we unravel a novel regulatory mechanism at *EIN2*, a multifunctional component of a critical stress regulatory pathway in plants. We discovered that the *ein6-1* mutants carry mutations in two genes that encode the H3K27me3 demethylase REF6 (EIN6) as well as a crucial subunit of the INO80 chromatin remodeling complex (EEN). Remarkably only the presence of both mutations led to the ectopic accumulation of H3K27me3 and H2A.Z at *EIN2*. The resulting repressive chromatin state compromises *EIN2* mRNA expression leading to a strongly reduced ET-induced accumulation of EIN3 protein and ultimately to a failure of the transcriptional response.

H3K27me3 and H2A.Z strongly co-associate in *Arabidopsis* which is not surprising since the majority of *Arabidopsis* genes are marked with H2A.Z (*Dong et al., 2012*; *Carter et al., 2018*; *Gómez-Zambrano et al., 2019*). Interestingly, they mutually reinforce their deposition through the concerted action of the methyltransferase CURLY LEAF (CLF) and the ATP-dependent chromatin remodeler PIE1 and PICKLE (PKL) (*Carter et al., 2018*). PIE1-dependent H2A.Z incorporation promotes the deposition of H3K27me3 which is retained after DNA replication by PKL (*Carter et al., 2018*). This scenario of a mutual reinforcement can explain the accumulation of H3K27me3 and H2A.Z in the *ein6-1 een-1* double mutant background. It's possible that non-evicted H2A.Z at *EIN2* initially leads to increased deposition of H3K27me3. Since H3K27me3 demethylation and H2A.Z eviction are both functionally impaired in *ein6-1 een-1* mutants, H3K27me3 and H2A.Z can continuously reinforce their deposition, which ultimately leads to the observed gene body-wide spreading of both marks at *EIN2*. Another interesting aspect arose from a recent study showing that monoubiquitination of H2A.Z confers H2A.Z-mediated transcriptional repression (*Gómez-Zambrano et al., 2019*). Although our anti-H2Aub antibody cannot distinguish between monoubiquitinated H2A and monoubiquitinated H2A.Z, we speculate that the observed increase of H2Aub at the gene body of *EIN2* in *ein6-1 een-1* mutants is predominantly monoubiquitinated H2A.Z. These findings suggest a scenario of mutually reinforced H3K27me3/H2A.Z deposition potentially combined with the monoubiquitination of H2A.Z that ultimately represses *EIN2* expression.

However, it's unclear why the increased mutually enforced deposition of H3K27me3 and H2A.Z can only be observed in the double mutant background. Although the functional loss of EIN6 (REF6) as the major *Arabidopsis* H3K27me3 demethylase has a profound impact on the H3K27me3 landscape in *Arabidopsis* (*Lu et al., 2011*), *EIN2* is only marginally affected. A molecular hallmark of *ein6* (*ref6*) mutants is the ectopic accumulation of H3K27me3 at thousands of genes (2369 Group I genes) (*Lu et al., 2011*). Our in-depth investigation of the *ein6-1 een-1* epigenome revealed a new class of genes (Group II genes), such as *EIN2,* that ectopically accumulates H3K27me3 in the *ein6-1* (*ref6-1*) mutant background but only when INO80 is simultaneously lost (*ref6-1 ino80-1*) or the INO80 complex activity is simultaneously impaired (*ein6-1 een-1*, *ref6-1 een-2*, *ref6-1 arp5-1*). These findings

indicate an EIN6 (REF6)-independent INO80-mediated mechanism that specifically prevents H3K27me3 hyper-accumulation at *EIN2* in the absence of EIN6.

The precise mechanism behind the observed functional redundancy of EIN6 (REF6) and the INO80 complex in regulating H3K27me3 occupancy at *EIN2* remains unclear. Regulatory interactions between H3K27me3 demethylases and SWI/SNF-type chromatin remodelers are described in various organisms (*Li et al., 2016*; *Miller et al., 2010*). The mouse H3K27me3 demethylases Jmjd3 and UTX interact with T-box transcription factors and a Brg1-containing SWI/SNF remodeling complex to activate gene expression in a H3K27me3 demethylase-independent manner (*Miller et al., 2010*). In *Arabidopsis*, EIN6 (REF6) physically interacts with BRAHMA (BRM) and is crucially involved in recruiting BRM to its target genes (*Li et al., 2016*). However, in contrast to *ein6-1 een-1* double mutants, *brm-1 ref6-1* double mutants show no additional ectopic H3K27me3 domains (*Li et al., 2016*).

The most prominent function of the INO80 chromatin remodeling complex is the eviction of the histone variant H2A.Z (*Papamichos-Chronakis et al., 2011*; *Alatwi and Downs, 2015*). This is corroborated by our findings of impaired ET-induced H2A.Z eviction dynamics in *een-1* and *ino80-8* mutants, as well as the H2A.Z accumulation at *EIN2* in *ein6-1 een-1* double mutants. However, in the absence of a stimulus or an *ein6* (*ref6*) background mutation, single mutants of the *Arabidopsis* INO80 complex (*een-1*, *een-2*, *arp5-1*, *ino80-1*, *ino80-8*) have no impact on global H2A.Z occupancy which in line with a recent report in yeast (*Jeronimo et al., 2015*). This strongly suggests that, like in mammals, there is an INO80-independent H2A.Z removal mechanism in *Arabidopsis* (*Obri et al., 2014*). The INO80 complex is crucial for pluripotency of embryonic stem cells (ESCs) (*Wang et al., 2014*). Silencing of INO80 leads to a breakdown of the ESC-specific chromatin structure that is accompanied by the accumulation of H3K27me3 at INO80 target sites (*Wang et al., 2014*). Based on this and our findings, it is conceivable that the INO80 complex can prevent H3K27me3 accumulation by antagonizing PRC2-mediated H3K27me3 deposition through direct or indirect blocking of the access of PRC2 to chromatin.

Loss of EIN6 (REF6) also has no effect on the global H2A.Z distribution, but leads to an ectopic H2A.Z accumulation at *EIN2* in the *een* mutant. This suggests that EIN6 can not only control chromatin structure through H3K27me3 demethylation, but also through the regulation of H2A.Z occupancy, although in a very specific genomic context. This is supported by our finding that the hyposensitive triple response phenotype of *ref6-1 een-2* mutants can be suppressed by the loss of SWR1-deposited H2A.Z in *ref6-1 een-2 pie1-1* triple mutants. Similar to the INO80-mediated restriction of PRC2 chromatin access, EIN6 (REF6) might control H2A.Z occupancy through blocking SWR1-mediated deposition of H2A.Z. We speculate that the observed functional duality of EIN6 (REF6) and the INO80 complex in antagonizing SWR1-mediated H2A.Z incorporation and PRC2-mediated H3K27 tri-methylation represents a double safeguard mechanism that ensures a permissive chromatin structure at *EIN2* to ultimately safeguard its expression (*Figure 5C*).

The question of why a gene like *EIN2* is controlled by this pivotal chromatin modifier equilibrium is not fully answered. Since *EIN2* is not the only gene that is directly targeted by EIN6 and EEN, other specificity determinants must exist. The majority of chromatin regulation at *EIN2* takes place in the 5'UTR intron region and our functional analysis of this region indicates the presence of multiple regulatory elements. Thus, we believe there is likely a yet unknown transcription factor that is required to establish the EIN6/INO80-dependent double safeguard mechanism at *EIN2* (*Figure 4C*). Besides safeguarding *EIN2* expression, it's also plausible that the EIN6/INO80-mediated double safeguard is actively turned off to allow stable repression of *EIN2* in a specific cell type and/or at a specific developmental stage. Histone signatures are mitotically stable and potentially echo prior or missing regulatory steps (*Wang and Higgins, 2013*). In addition, with our finding that dry *ein6-1 een-1* seeds already display the aberrant chromatin signature at *EIN2*, it can be envisioned that the EIN6/INO80-mediated regulation already takes place during embryogenesis. This might also provide an explanation for the relatively weak binding of EIN6 and EEN to *EIN2*. Since we were able to capture strong EIN6 and EEN binding at numerous loci, technical reasons for weak binding such as insufficient crosslinking to the DNA are less likely. It is also plausible that the number of cells with EIN6/EEN-binding at *EIN2* is rather small or that DNA residency times and chromatin remodeler compositions may vary between genes, thus potentially affecting the outcome of our ChIP-seq experiments. Given EIN2's multifunctional role in controlling many agronomically important processes (*Johnson and Ecker, 1998*), we propose that this newly discovered mode of regulation at

*EIN2*, not only fine-tunes its expression, but may also provide an efficient switch to repress gene expression.

## Materials and methods

**Key resources table**

| Reagent type (species) or resource | Designation | Source or reference | Identifiers | Additional information |
|---|---|---|---|---|
| Strain, strain background (*A. thaliana*) | *ein6-1 een-1* | *Roman et al. (1995)* | | |
| Strain, strain background (*A. thaliana*) | *ein6-1* | This study | | Materials and methods subsection: Plant material and growth conditions |
| Strain, strain background (*A. thaliana*) | *een-1* | This study | | Materials and methods subsection: Plant material and growth conditions |
| Strain, strain background (*A. thaliana*) | *ein2-45* | *Beaudoin et al. (2000)* | | |
| Strain, strain background (*A. thaliana*) | *ein2-5* | *Alonso et al. (1999)* | | |
| Strain, strain background (*A. thaliana*) | *ref6-1* | *Noh et al. (2004)* | | |
| Strain, strain background (*A. thaliana*) | *een-2* | This study | SALKseq_129237 | Materials and methods subsection: Plant material and growth conditions |
| Strain, strain background (*A. thaliana*) | *arp5-1* | *Kandasamy et al. (2009)* | | |
| Strain, strain background (*A. thaliana*) | *ino80-1* | *Fritsch et al. (2004)* | | |
| Strain, strain background (*A. thaliana*) | *ino80-8* | This study | SALKseq_041809 | Materials and methods subsection: Plant material and growth conditions |
| Strain, strain background (*A. thaliana*) | *pie1-1* | *Noh and Amasino (2003)* | | |
| Strain, strain background (*A. thaliana*) | *ref6-1 een-2* | This study | | Materials and methods subsection: Plant material and growth conditions |
| Strain, strain background (*A. thaliana*) | *ref6-1 arp5-1* | This study | | Materials and methods subsection: Plant material and growth conditions |
| Strain, strain background (*A. thaliana*) | *ref6-1 ino80-1* | This study | | Materials and methods subsection: Plant material and growth conditions |
| Strain, strain background (*A. thaliana*) | *pie1-1 ref6-1 een-2* | This study | | Materials and methods subsection: Plant material and growth conditions |
| Strain, strain background (*A. thaliana*) | Col-0 *HTA11:HTA11:GFP* | *Kumar and Wigge (2010)* | | |

*Continued on next page*

Continued

| Reagent type (species) or resource | Designation | Source or reference | Identifiers | Additional information |
|---|---|---|---|---|
| Strain, strain background (*A. thaliana*) | *Ler 35S:EIN6-FLAG* | This study | | Materials and methods subsection: Vectors construction and plant transformation conditions |
| Strain, strain background (*A. thaliana*) | *Col-0 35S:FLAG-EEN* | This study | | Materials and methods subsection: Vectors construction and plant transformation conditions |
| Strain, strain background (*A. thaliana*) | *Col-0 35S:GFP-EEN* | This study | | Materials and methods subsection: Vectors construction and plant transformation conditions |
| Strain, strain background (*A. thaliana*) | *ein6-1 een-1 EIN6:EIN6* | This study | | Materials and methods subsection: Vectors construction and plant transformation conditions |
| Strain, strain background (*A. thaliana*) | *ein6-1 een-1 EEN:EEN* | This study | | Materials and methods subsection: Vectors construction and plant transformation conditions |
| Antibody | Rabbit polyclonal anti-histone H3K27me3 | Active Motif | Cat# 39156, RRID:AB_2636821 | 5 µl Materials and methods subsection: ChIP-seq |
| Antibody | Rabbit polyclonal anti - Htz1/Histone H2A.Z | Active Motif | Cat# 39647 RRID: AB_2793289 | 10 µl Materials and methods subsection: ChIP-seq |
| Antibody | Rabbit monoclonal anti trimethyl-Histone H3 (Lys4) | Millipore Sigma | Cat#: 04–745 RRID:AB_1163444 | 4 µl Materials and methods subsection: ChIP-seq |
| Antibody | Rabbit monoclonal anti-ubiquityl-Histone H2A (Lys119) | Cell Signaling Technology | Cat#: 8240 RRID: AB_10891618 | 10 µl Materials and methods subsection: ChIP-seq |
| Antibody | Mouse monoclonal anti-FLAG M2 | Millipore Sigma | Cat#: F1804 RRID: AB_262044 | 5 µl Materials and methods subsection: ChIP-seq |
| Antibody | Mouse monoclonal anti-GFP | Millipore Sigma | Cat#:11814460001 RRID: AB_390913) | 5 µl Materials and methods subsection: ChIP-seq |
| Antibody | ChromPure Mouse IgG, whole molecule | Jackson ImmunoResearch | Cat#:015-000-003 RRID: AB_2337188 | 2 µl Materials and methods subsection: ChIP-seq |
| Antibody | Mouse monoclonal anti-HaloTag | Promega | Cat#: G9211 | 1:1000 Materials and methods subsection: Immunoblot analysis and in vitro pull-down assays |
| Antibody | Rabbit polyclonal anti-Histone H3 | Abcam | Cat#: ab1791 RRID:AB_302613) | 1:1000 Materials and methods subsection: Immunoblot analysis and in vitro pull-down assays |

*Continued on next page*

*Continued*

| Reagent type (species) or resource | Designation | Source or reference | Identifiers | Additional information |
|---|---|---|---|---|
| Antibody | Rabbit polyclonal anti-EIN3 | *Chang et al. (2013)* | | 1:500 Materials and methods subsection: Immunoblot analysis and in vitro pull-down assays |
| Antibody | Mouse polyclonal anti-EIN2 | *Qiao et al. (2012)* | | 1:500 Materials and methods subsection: Immunoblot analysis and in vitro pull-down assays |
| Chemical compound, drug | Protein G Dynabeads | Thermo Fisher Scientific | Cat. #: 10004D | 50 µl Materials and methods subsection: ChIP-seq |

## Plant material and growth conditions

The *ein6-1 een-1* mutant resulted from a genetic screen using fast neutron-mutagenized L*er* seeds as previously described (*Roman et al., 1995*). Both an *ein2* mutant allele in the L*er* background (*ein2-45*) and in the Col-0 background (*ein2-5*) were used in this study (*Beaudoin et al., 2000*; *Alonso et al., 1999*). The *ref6*, *ino80* and *arp5* mutants used in this study were *ref6-1*, *ino80-1* and *arp5-1* (*Noh et al., 2004*; *Fritsch et al., 2004*; *Kandasamy et al., 2009*). In addition, we isolated a new *ino80* allele (SALKseq_041809) which now refers to *ino80-8*. The newly isolated *een-2* T-DNA mutant (SALKseq_129237) as well as the *ino80-8* mutant was obtained from the Arabidopsis Biological Resource Center (ABRC). The *pie1* mutant that was used in this study is *FRIGIDA* (*FRI*) *pie1-1* (*Noh and Amasino, 2003*). Transgenic Col-0 *HTA11:GFP* plants express a previously described functional *HTA11:GFP* cassette (*Kumar and Wigge, 2010*). Seedlings were grown on Linsmaier and Skoog (LS) medium containing 0.8% agar and 1% sucrose. For the triple response assay, surface-sterilized wild-type (Col-0, L*er*) and mutant seeds were grown on agar plates containing LS medium supplemented with or without 10 µM 1-aminocyclopropane-1-carboxylic acid (ACC) for three days in the dark. For the induction with ET, seedlings were grown for three days in the dark in the presence of hydrocarbon free air and subsequently treated with ethylene gas for 4 hr. Flower tissue for mass spectrometry analyses was collected from 4 to 6 weeks old plants that were grown under controlled environmental conditions (21/19°C, 16-h-light/8-h-dark cycle). *ref6-1 een-2*, *ref6-1 ino80-1* and *ref6-1 arp5-1* double mutants were generated by crossing of *ref6-1* single mutant pollen into the respective mutants. The *ref6-1 een-2 pie1-1* triple mutant originates from a cross of the *ref6-1 een-2* double mutant with pollen of *pie1-1* mutants. The F1-progeny was allowed to self-fertilize and the resulting F2 generation was screened for homozygosity using PCR (for primer sequences see *Supplementary file 9*).

## Mutant mapping and characterization of mutations

An *ein6-1 een-1* mutant was crossed into the Col-0 background and pooled DNA of approximately 350 ET-insensitive F2 seedlings was used for SHOREmap analysis (*Schneeberger et al., 2009*). ET-insensitivity was determined with the triple response assay and genomic DNA was extracted with the DNeasy Plant Mini Kit (69104, Qiagen). The 7 bp deletion in the *EIN6* (*REF6*) gene was discovered through PCR amplification of the *EIN6* (*REF6*) gene from *ein6-1* mutants and subsequent Sanger sequencing of the amplicon. The inversion event that led to the fusion of the *EEN* gene with the *AQI* gene was initially detected with Thermal Asymmetric Interlaced (TAIL)-PCR (*Singer and Burke, 2003*). Three primers (TAIL1-3) that specifically anneal to the 5'UTR region of *EEN* and six degenerated primers (AD1-6) were used (*Supplementary file 9*). Two amplicons that show the expected size difference were extracted using the QIAquick Gel Extraction Kit (28704, Qiagen). The DNA sequences of the purified amplicons were determined by Sanger sequencing. The *EEN-AQI* fusion event was further characterized by the amplification and subsequent sequencing of the chimeric region using genomic DNA and cDNA as a template (for primer sequences see *Supplementary file 9*). Visualization of the inversion region was achieved with optical mapping of

ultra-long DNA molecules of Ler and ein6-1 een-1 seedlings using the Bionano Genomics Irys instrument. For the optical mapping, genomic DNA was extracted from nuclei and processed as previously described (*Kaufmann et al., 2010*; *Kawakatsu et al., 2016*).

## Vectors construction and plant transformation

Gateway cloning and TOPO cloning (Thermo Fisher Scientific) was used to generate the majority of vectors in this study. For complementation analyses, genomic fragments of *EIN6* (*REF6*, *At3g48430*) and *EEN* (*At4g38495*) including a 2 kb promoter region but without the stop codon were amplified using genomic DNA from Ler plants as a template. Amplified fragments were inserted into pENTR/D-TOPO via TOPO cloning and subsequently cloned via LR reaction into the binary vector pEarly-Gate302 (https://www.arabidopsis.org/abrc/catalog/vector_1.html). For the generation of transgenic Cauliflower mosaic virus *35S* promoter-driven *EEN* (*35S:EEN*) plants, the coding sequences of *EIN6* (*REF6*) and *EEN* were first amplified from Ler cDNA, introduced into pENTR/D-TOPO and subsequently cloned via LR reaction into the binary vector pEarleyGate202 (https://www.arabidopsis.org/abrc/catalog/vector_1.html). The binary vector pMDC163 (https://www.arabidopsis.org/abrc/catalog/vector_1.html) containing a *gusA* gene as a reporter was used for the *EEN* expression analysis, as well as for the stable and transient *EIN2* promoter/5'UTR intron analyses. An approximately 2 kb promoter fragment of *EEN* and the respective *EIN2* promoter/5'UTR fragments were amplified from genomic Ler DNA, cloned into pENTR/D-TOPO and inserted by LR reaction into pMDC163. For the identification of the subcellular localization of GFP-tagged EEN, pENTR/D-TOPO-*EEN* was used to introduce the cDNA of *EEN* via LR reaction into pGWB6 (http://shimane-u.org/nakagawa/gbv.html). For in vitro pulldown experiments, pENTR/D-TOPO-*EEN* was used to introduce the cDNA of *EEN* via LR reaction into a modified version of the pTNT plasmid (L5610, Promega) containing a N-terminal FLAG-tag (*Nito et al., 2013*). The *ARP5* cDNA was cloned into pDONR221 (12536017, Thermo Fisher Scientific) via BP reaction and subsequently introduced into pIX-HALO (https://www.arabidopsis.org/servlet/TairObject?id=1001200298&type=vector) via LR reaction. For DAP-seq assays, the C-terminus of *EIN6* (156aa) were cloned into pENTR/D-TOPO and subsequently introduced via LR reaction into pIX-HALO. For the overexpression of N-terminal FLAG-tagged full-length and truncated EIN6 (*35S:EIN6, 35S:EIN6ΔZNF*), the respective coding sequences without a stop codon were amplified from Ler cDNA. Restriction sites for KpnI and SalI as well as a FLAG-tag with stop codon were added through amplification. KpnI/SalI-digested amplicons were subsequently ligated into the binary vector pCHF3. Binary plasmids were electroporated into *Agrobacterium tumefaciens* strain GV3101 and subsequently transformed into the respective *Arabidopsis* plants using the floral dip method (*Clough and Bent, 1998*). Primers used to generate all described constructs are listed in *Supplementary file 9*.

## ChIP-seq

ChIP-seq experiments using three-day-old etiolated seedlings were performed as previously described (*Kaufmann et al., 2010*) with minor modifications. ChIP-seq assays were conducted with antibodies against H3K27me3 (39156, Active Motif), H2A.Z (39647, Active Motif), H3K4me3 (04–745, Millipore Sigma), H2Aub (8240, Cell Signaling Technology), FLAG (F1804, Millipore Sigma) and GFP (11814460001, Millipore Sigma). Mouse IgG (015-000-003, Jackson ImmunoResearch) served as the negative control. Protein G Dynabeads (50 μl, 10004D, Thermo Fisher Scientific) were coupled for 4–6 hr with the respective antibodies or IgG and subsequently incubated overnight with equal amounts of sonicated chromatin. Beads were washed successively with high salt buffer (50 mM Tris HCl pH 7.4, 150 mM NaCl, 2 mM EDTA, 0.5% Triton X-100), low salt (50 mM Tris HCl pH 7.4, 500 mM NaCl, 2 mM EDTA, 0.5% Triton X-100) and wash buffer (50 mM Tris HCl pH 7.4, 50 mM NaCl, 2 mM EDTA) before de-crosslinking, proteinase K digestion and DNA precipitation. Libraries were sequenced on the Illumina HiSeq 2500 and HiSeq 4000 Sequencing system. Sequencing reads were aligned to the TAIR10 genome assembly using Bowtie2 (*Langmead, 2010*).

## DAP-seq

DAP-seq assays were carried as previously described (*O'Malley et al., 2016*) using the recombinantly expressed C-terminus of EIN6 (REF6) and genomic Col-0 DNA.

## RNA-seq

Three-day-old etiolated seedlings were used for expression analyses. Total RNA was extracted with the RNeasy Plant Mini Kit (74903, Qiagen). cDNA library preparation and subsequent single read sequencing was carried as previously described (*Song et al., 2016*). Sequencing reads were aligned to the TAIR10 genome assembly using the STAR software (version STAR_2.6.0 c) (*Dobin et al., 2013*).

## MethylC-seq

MethylC-seq assays were conducted as previously described (*Lister et al., 2011*). Differentially methylated regions (DMRs) in L*er*, *ein6-1*, *een-1* and *ein6-1 een-1* seedlings were identified with the methylpy software (*Schultz et al., 2015*).

## Data analysis

Significant enrichments of histone modifications, histone variants, EIN6 and EEN were identified with the SICER software (*Zang et al., 2009*) using the TAIR10 genome assembly. The Intersect tool from BEDtools (*Quinlan and Hall, 2010*) was used to identify the genes in the ChIP-seq datasets that are most proximal to the discovered binding sites. The Genome wide Event finding and Motif discovery (GEM) tool (*Guo et al., 2012*) was used to identify the peak summit regions in ChIP/DAP-seq data-derived binding sites of EIN6. The MEME-ChIP analysis tool (*Machanick and Bailey, 2011*) was then used to identify binding motifs within the top 1000 summit regions of three merged EIN6 ChIP-seq experiments and one DAP-seq experiment. Gene ontology (GO) enrichment analysis was carried out with DAVID (*Dennis et al., 2003*). For the analysis of genomic distributions within EEN ChIP-seq data, all binding sites were first identified with SICER (version 1.1) and then subjected to the *cis*-regulatory element annotation system (CEAS) tool (*Shin et al., 2009*). To generate aggregated profiles, heatmaps and correlation analyses of ChIP-seq data, Deeptools (*Ramírez et al., 2014*) was employed. For the validation of our H2A.Z antibody, three published H2A.Z ChIP-seq datasets were downloaded (*Carter et al., 2018*) (GSE103361; sample GSM2769481), *Wollmann et al., 2017* (GSE96834; sample GSM2544791), *Coleman-Derr and Zilberman, 2012* (GSE39045; sample GSM954614)). multiBamSummary and plotCorrelation from Deeptools was used to generated the Spearman's correlation plot. A total of 59814 2 kb bins were found. For the quantification of occupancy of the various histone modifications or of H2A.Z at the *EIN2* gene, the bigWigAverageOverBed tool executable from the UCSC genome browser was used (*Kent et al., 2002*). Quantification was done for the 5'UTR intron region (chr5, 787428–788588), gene body region (chr5, 788589–793066) and 3'UTR intron region (chr5, 793067–793356). SAMtools (*Li et al., 2009*) was used to merge biological ChIP-seq replicates. For overlap analysis (http://bioinformatics.psb.ugent.be/webtools/Venn/), pairwise comparisons of ChIP-seq samples (*ein6-1 een-1* >*ein6-1* (H3K27me3, 2-fold), *ein6-1 een-1* >*een-1* (H2A.Z, 1,5-fold), *ein6-1 een-1* >L*er* (H2Aub, 2-fold), L*er* >*ein6-1 een-1* (H3K4me3, 1.5-fold)), using SICER were conducted to discover genes with differential enrichment of the respective modification or of H2A.Z in *ein6-1 een-1* mutants. DMRs with ten or more methylated cytosines in at least one genotype were included into the overlap analysis as well. For the identification of genes that show a robust ET-induced eviction of H2A.Z, we first employed SICER (≥1.3 fold SICER comparison, air vs 4 hr ET) to identify all genes that show a significant eviction of H2A.Z after ET-treatment in L*er* (two replicates) and Col-0 (one replicate). 1076 genes were identified in all replicates and were used to visualize ET-induced H2A.Z eviction dynamics in *een-1* and *ino80-8* mutants.

The RSEM software package (version 1.3.0) was used to quantify transcripts in the RNA-seq experiments (*Li and Dewey, 2011*). To identify differentially expressed genes in the RNA-seq datasets, the edgeR package (*Robinson et al., 2010*) was used. For the visualization of the transcriptional ET response in L*er*, *ein6-1*, *een-1*, *ein6-1 een-1* and *ein2-45* seedlings *via* a heatmap, all genes with a significant $\log_2$ fold expression change in L*er* seedlings in response to 4 hr of ET treatment (1803 induced, 2598 repressed) were selected. To display the transcriptomic similarities between *ino80-8*, *een-2* and *arp5-1* mutants, INO80-dependent genes were first identified by a direct comparison of the Col-0 with the *ino80-8* transcriptome (787 genes up, 1079 genes down) and their expression relative to Col-0 was then visualized with a heatmap. All sequencing data were visualized with the AnnoJ genome browser (*Lister et al., 2008*). ClustalW2 (*Larkin et al., 2007*) was used for multiple amino acid sequence alignment of EEN (Accession AEE86934.1), IES6 (Accession P32617.1)

and human IES6 (hIES6) (Accession AAH39404.1). For the interrogation of TFs that can bind to the 5'UTR region of *EIN2*, the Plant Cistrome Database (*O'Malley et al., 2016*) (http://neomorph.salk.edu/dap_web/pages/index.php) was used.

## Immunoblot analysis and in vitro pull-down assays

Nuclear proteins of untreated and ET-treated *Arabidopsis* seedling tissue were extracted as previously described (*Kaufmann et al., 2010*). For immunoblots analysis, nuclear extracts were boiled in 1X LDS sample buffer (NP0008, Thermo Fisher Scientific) and extracted proteins were subsequently separated by SDS-PAGE. The TnT Quick Coupled Transcription/Translation System (L1170, Promega) was used for in vitro pull-down assays. in vitro translated FLAG-tagged EEN was incubated with Anti-FLAG M2 Magnetic Beads (M8823, Millipore Sigma) for 1 hr and subsequently washed with WP buffer (50 mM Tris HCl pH 7.5, 100 mM NaCl, 1 mM EDTA, 1% DMSO, 2 mM TCEP pH 7.5, 0.1 % NP-40). After washing, the beads were incubated with in vitro translated Halo-tagged ARP5 for additional 2 hr. After three washes with WP buffer, beads were boiled in 1X LDS sample buffer and the supernatant was subjected to immunoblot analysis. Specificities of the used antibodies against EIN2 and EIN3 were previously tested (*Qiao et al., 2012*; *Chang et al., 2013*; *Kandasamy et al., 2009*). Antibodies against FLAG, HA, GFP (F1804, 11867423001, 11814460001, Millipore Sigma) and Halo (G9211, Promega) were used for the respective immunoblot analysis. An anti-histone H3 antibody (ab1791, Abcam) was used to control for equal loading.

## Fluorescence microscopy, histochemical GUS staining, and quantitative GUS assays

Fluorescence microscopy was performed with 10-day-old Col-0 *35S:GFP-EEN* seedlings to identify the subcellular localization of EEN. Images were taken with a Zeiss LSM 710 Confocal Microscope. GUS staining was carried out as described before (*Willige et al., 2011*). Transgenic *Arabidopsis* seedlings either harboring an *EEN:GUS, EIN2:GUS or EIN2ΔUI:GUS* cassette were incubated in the GUS staining solution for 12 hr. For quantitative GUS assays, *Nicotiana benthamiana* leafs were infiltrated with the *Agrobacterium tumefaciens* strain GV3101 harboring the respective binary plasmids with an incubation time of 48 hr as previously reported (*Bürger et al., 2017*). Fluorometric quantification of GUS reporter activity was carried out as previously described (*Zander et al., 2014*).

## Mass spectrometry analysis

Nuclei were extracted from flowers of Col-0 *35S:EEN-FLAG* as previously described (*Kaufmann et al., 2010*). Nuclear extracts of non-transgenic Col-0 served as a control. Anti-FLAG M2 Magnetic Beads (M8823, Millipore Sigma) were washed three times with Sonic buffer before they were incubated for 2 hr with equal amounts of nuclear protein. After three additional washes with Sonic buffer, beads were boiled in 1X LDS sample buffer and the supernatants were subjected to mass spectrometry analysis. Samples were precipitated by methanol/chloroform first and the dried pellets were dissolved in 8 M urea/100 mM TEAB, pH 8.5. Proteins were reduced with 5 mM Tris (2-carboxyethyl) phosphine hydrochloride (TCEP) and alkylated with 10 mM chloroacetamide. Proteins were digested overnight at 37 ° C in 2 M urea/100 mM TEAB, pH 8.5, with trypsin. Digestion was quenched with formic acid (5% final concentration). The digested samples were analyzed on a Fusion Orbitrap Tribrid mass spectrometer (IQLAAEGAAPFADBMBCX, Thermo Fisher Scientific). The digest was injected directly onto a 30 cm, 75 um ID column packed with BEH 1.7 um C18 resin (186002350, Waters). Samples were separated at a flow rate of 300 nl/min on a nLC 1000 liquid chromatograph (LC120, Thermo Fisher Scientific). Buffer A and B were 0.1% formic acid in water and 0.1% formic acid in 90% acetonitrile, respectively. A gradient of 1–25% B over 90 min, an increase to 40% B over 30 min, an increase to 90% B over 10 min and held at 90% B for a final 10 min of washing was used for 140 min total run time. The column was re-equilibrated with 20 ul of buffer A prior to the injection of samples. Peptides were eluted directly from the tip of the column and nanosprayed directly into the mass spectrometer by application of 2.5 kV voltage at the back of the column. The Orbitrap Fusion was operated in a data-dependent mode. Full MS scans were collected in the Orbitrap at 120K resolution with a mass range of 400 to 1600 m/z and an AGC target of $5e^5$. The cycle time was set to 3 s, and within this 3 s the most abundant ions per scan were selected for CID MS/MS in the ion trap with an AGC target of $1e^4$ and minimum intensity of 5000. Maximum fill times

were set to 50 ms and 100 ms for MS and MS/MS scans respectively. Quadrupole isolation at 1.6 m/z was used, monoisotopic precursor selection was enabled and dynamic exclusion was used with exclusion duration of 5 s. Protein and peptide identification were done with Integrated Proteomics Pipeline – IP2 (Integrated Proteomics Applications). Tandem mass spectra were extracted from raw files using RawConverter (*He et al., 2015*) and searched with ProLuCID (*Xu et al., 2015*; *Peng et al., 2003*) against the TAIR database with reversed sequences. The search space included all fully-tryptic and half-tryptic peptide candidates. Carbamidomethylation on cysteine was considered as a static modification. Data was searched with 50 ppm precursor ion tolerance and 600 ppm fragment ion tolerance. Identified proteins were filtered to using DTASelect (*Tabb et al., 2002*) and utilizing a target-decoy database search strategy to control the false discovery rate to 1% at the protein level (*Peng et al., 2003*).

## cDNA synthesis and RT-PCR

Total RNA from L*er* and *een-1* seedlings was extracted using the RNeasy Plant Mini Kit (74903, Qiagen) to investigate the *EEN-AQI* fusion event. Extracted RNA was reverse transcribed into cDNA using the High-Capacity cDNA Reverse Transcription Kit (4368814, Thermo Fisher Scientific). RT-PCR analysis of the chimeric *EEN-AQI* cDNA was carried with the Phusion High-Fidelity DNA Polymerase (M0530S, New England Biolabs). Primers used to amplify and quantify the respective cDNA levels are indicated in *Supplementary file 9*.

## Accession numbers

Sequence data can be downloaded from GEO (accession GSE122314). Visualized sequencing data can be found under http://neomorph.salk.edu/ein6een.php.

## Acknowledgements

We thank Shao-shan Carol Huang for providing computational assistance, Tsegaye Dabi for technical assistance, and James Moresco and Jolene Diedrich for technical support. We also thank Adam Seluzicki for useful comments on the manuscript and Hong Qiao for sharing the EIN2 antibody. Susan M Gasser, Richard Meagher and Philip A Wigge for kindly providing *ino80-1*, *arp5-1* and Col-0 *HTA11:GFP* seeds, respectively. This work was supported by the Mass Spectrometry Core of the Salk Institute with funding from NIH-NCI CCSG (P30 014195) and the Helmsley Center for Genomic Medicine. MZ was supported by the Salk Pioneer Postdoctoral Endowment Fund as well as by a Deutsche Forschungsgemeinschaft (DFG) research fellowship (Za-730/1–1). BCW was supported by the Salk Pioneer Postdoctoral Endowment Fund and the Human Frontier Science Program (LT000222/2013 L). MGL was supported by by an EU Marie Curie FP7 International Outgoing Fellowship (252475). This work was supported by grants from the National Science Foundation (NSF) (MCB-1024999), the Division of Chemical Sciences, Geosciences, and Biosciences, Office of Basic Energy Sciences of the US Department of Energy (DE-FG02-04ER15517), the Gordon and Betty Moore Foundation (GBMF3034) and National Institute of Health (NIH) grant 5R35 GM122604 (to JC). JC and JRE are Investigators of the Howard Hughes Medical Institute.

## Additional information

### Funding

| Funder | Grant reference number | Author |
|---|---|---|
| National Science Foundation | MCB-1024999 | Joseph R Ecker |
| Gordon and Betty Moore Foundation | GBMF3034 | Joseph R Ecker |
| Howard Hughes Medical Institute | | Joseph R Ecker |
| Deutsche Forschungsgemeinschaft | Za-730/1-1 | Joanne Chory Joseph R Ecker |

| | | |
|---|---|---|
| Salk Pioneer Postdoctoral Endowment Fund | | Mark Zander |
| Human Frontier Science Program | LT000222/2013-L | Mark Zander |
| EU 7th Framework Programme | EU Marie Curie FP7 International Outgoing Fellowship: 252475 | Björn C Willige |
| National Institutes of Health | 5R35 GM122604 | Mathew G. Lewsey |
| U.S. Department of Energy | DOE FG02 04ER15517 | Joanne Chory |

The funders had no role in study design, data collection and interpretation, or the decision to submit the work for publication.

## Author contributions

Mark Zander, Conceptualization, Data curation, Validation, Investigation, Visualization, Methodology, Writing—original draft; Björn C Willige, Anna Stepanova, Robert J Schmitz, Mathew G Lewsey, Investigation, Writing—review and editing; Yupeng He, Formal analysis, Validation, Visualization; Thu A Nguyen, Amber E Langford, Ramlah Nehring, Elizabeth Howell, Robert McGrath, Investigation; Anna Bartlett, Rosa Castanon, Joseph R Nery, Investigation, Methodology; Huaming Chen, Data curation; Zhuzhu Zhang, Florian Jupe, Methodology; Joanne Chory, Writing—review and editing; Joseph R Ecker, Conceptualization, Supervision, Funding acquisition, Project administration, Writing—review and editing

## Author ORCIDs

Mark Zander https://orcid.org/0000-0001-8643-1407
Björn C Willige https://orcid.org/0000-0003-4275-5826
Yupeng He https://orcid.org/0000-0001-9313-614X
Amber E Langford https://orcid.org/0000-0001-9573-9905
Robert McGrath https://orcid.org/0000-0002-4002-1522
Huaming Chen https://orcid.org/0000-0001-5289-7882
Florian Jupe https://orcid.org/0000-0001-5741-4931
Robert J Schmitz https://orcid.org/0000-0001-7538-6663
Mathew G Lewsey https://orcid.org/0000-0002-2631-4337
Joseph R Ecker https://orcid.org/0000-0001-5799-5895

## Decision letter and Author response

Decision letter https://doi.org/10.7554/eLife.47835.026
Author response https://doi.org/10.7554/eLife.47835.027

## Additional files

### Supplementary files

• Supplementary file 1. Differentially regulated genes in L*er*, *ein6-1*, *een-1*, *ein6-1 een-1* and *ein2-45* seedlings in response to ET. Tables displaying differentially regulated ET genes in L*er*, *ein6-1*, *een-1*, *ein6-1 een-1* and *ein2-45* seedlings. Differentially expressed genes were discovered with edgeR.
DOI: https://doi.org/10.7554/eLife.47835.013

• Supplementary file 2. Sequencing samples. Table displaying all generated sequencing samples.
DOI: https://doi.org/10.7554/eLife.47835.014

• Supplementary file 3. Group I and Group II genes. Tables displaying Group I and Group II genes which were identified by SICER through the comparison of *ein6-1* vs L*er* (Group I) and *ein6-1 een-1* vs *ein6-1* (Group II).
DOI: https://doi.org/10.7554/eLife.47835.015

• Supplementary file 4. Differentially enriched epigenome features in *ein6-1 een-1* mutants. Table displaying differentially enriched epigenome features in *ein6-1 een-1* mutants. Epigenome feature differences were determined with SICER (H2A.Z, H3K4me3 and H2Aub) and methylpy (5mC).
DOI: https://doi.org/10.7554/eLife.47835.016

• Supplementary file 5. Binding sites of EIN6, EIN6-ZNF and EEN. Tables display the binding sites of EIN6, EIN6-ZNF and EEN that were determined with SICER and GEM.
DOI: https://doi.org/10.7554/eLife.47835.017

• Supplementary file 6. Potential EEN interactors. Tables display the two IP mass spectrometry replicates using Col-0 *35S:EEN-FLAG* flowers (MS-IP I and MS IP II). Only potential interactors that show no spectral counts in the IgG sample were included.
DOI: https://doi.org/10.7554/eLife.47835.018

• Supplementary file 7. Differentially regulated genes in untreated *ino80-8*, *een-2* and *arp5-1* seedlings. Tables displaying differentially regulated genes in *ino80-8*, *een-2* and *arp5-1* seedlings. Differentially expressed genes were discovered with edgeR.
DOI: https://doi.org/10.7554/eLife.47835.019

• Supplementary file 8. Genes with a robust eviction of H2A.Z in response to ET. Tables displays genes that a robust ET-induced eviction of H2A.Z in L*er* and Col-0 (1.3 fold enrichment). Significant differential enrichment was determined with SICER. A list
DOI: https://doi.org/10.7554/eLife.47835.020

• Supplementary file 9. List of primers. Table displays all used primer in this study.
DOI: https://doi.org/10.7554/eLife.47835.021

• Transparent reporting form
DOI: https://doi.org/10.7554/eLife.47835.022

## Data availability

Sequence data have been deposited in GEO under accession GSE122314. An overview of all sequenced data is given in Supplementary file 2. Visualized sequencing data can be found under http://neomorph.salk.edu/ein6een.php.

The following dataset was generated:

| Author(s) | Year | Dataset title | Dataset URL | Database and Identifier |
|---|---|---|---|---|
| Zander M, Willige BC, He Y, Nguyen TA, Langford AE, Nehring R, Howell E, McGrath R, Bartlett A, Castanon R, Nery JR, Chen H, Zhang Z, Jupe F, Lewsey MG, Stepanova AN, Schmitz RJ, Chory J, Ecker J | 2019 | Epigenetic Control of a Multifunctional Stress Regulator | https://www.ncbi.nlm.nih.gov/geo/query/acc.cgi?acc=GSE122314 | NCBI Gene Expression Omnibus, GSE122314 |

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
