## [Decision Letter]

Thank you for submitting your article "Epigenetic silencing of a multifunctional plant stress regulator" for consideration by *eLife*. Your article has been reviewed by three peer reviewers, including Daniel Zilberman as the Reviewing Editor, and the evaluation has been overseen by Christian Hardtke as the Senior Editor. The other reviewers have opted to remain anonymous.

The reviewers have discussed the reviews with one another and the Reviewing Editor has drafted this decision to help you prepare a revised submission.

Summary:

Zander et al. present a detailed analysis of the ethylene-insensitive *ein6-1* mutant, which, fascinatingly, turns out to carry mutations in both the REF6 H3K27me3 demethylase and a putative subunit (EEN) of the INO80 chromatin remodeling complex. Both mutations are required for the ethylene insensitivity phenotype, which apparently results from silencing of *EIN2* in the double mutant. The authors show that both H3K27me3 and H2A.Z accumulate at the *EIN2* locus in the double mutant, suggesting that REF6 and INO80 cooperate to prevent polycomb silencing of *EIN2*. Interestingly, they also present evidence that this silencing is initiated at an intron in the 5' UTR of the *EIN2* gene. Although the precise mechanism underlying this phenomenon remains to be elucidated, the results are very exciting in that they shed light on the control of a central player in the ethylene response and add further depth to the recently identified connection between H2A.Z deposition and Polycomb silencing.

Essential revisions:

1) The interaction between H2A.Z and H3K27me3 was recently described in Arabidopsis, as the authors note: "H2A.Z and H3K27me3 are functionally linked and both repress transcription in Arabidopsis (Carter et al., 2018, Coleman-Derr and Zilberman, 2012)." However, the authors do not mention the work by Carter et al. further, including in the discussion of how H3K27me3 and H2A.Z might interact (Discussion). Carter et al. show that H2A.Z deposition in gene bodies is dependent on the H3K27me3 methyltransferase CLF, and H3K27me3 is promoted by the H2A.Z deposition factor PIE1. They propose a model for mutual reinforcement, where H2A.Z incorporation promotes H3K27me3 and H3K27me3 stabilizes H2A.Z by fostering nucleosome retention. Importantly, the work by Carter et al. indicates that the relationship between H3K27me3 and H2A.Z is quite general and not confined to a few genes that include *EIN2*. The authors should discuss their findings in light of this work, including in the model in Figure 5C. The integration of work by Carter et al. should also clarify the model, as the "blocking" connectors from EIN6 to SWR1 and from INO80 to PRC2 are not apparently supported by sufficient data. The authors may also wish to cite Dong et al., 2012, which noted an enrichment of H2A.Z associated with H3K27me3.

2) The authors' results and those of Carter et al. indicate that K3K27me3 should be lost at *EIN2* in *ref6;een;pie1* triple mutants. The authors should evaluate this, as the loss of K3K27me3 at *EIN2* is an important prediction of their model.

3) The H2A.Z antibody used in this study is a commercial preparation raised against the full-length budding yeast H2A.Z protein. The WT and pie1 ChIP-seq data strongly suggest that this antibody recognizes Arabidopsis H2A.Z, but given the importance of the data for the paper's conclusions, validation of the antibody and the ChIP-seq data is important. At minimum, please include a comprehensive comparison of the H2A.Z ChIP-seq data to one or more of the publicly available Arabidopsis ChIP-seq datasets derived from well-validated H2A.Z antibodies. A western blot using the antibody on WT and pie1 chromatin fractions would also be helpful.

4) The reviewers expressed concerns about the re-ChIP data-set. Visualization of these data (as provided in the authors' link) indicates that they are not very convincing and that the IgG control is not clean. Since the data are not extensively integrated into the major conclusions, we suggest removing them. Alternatively, please include an analysis that demonstrates the validity of these data.

5) Related to point 4, the EEN-ChIP dataset did not show a convincing signal to noise ratio, despite the great number of peaks identified by SICER. In Figure 3A, the authors display a profile of enrichment over background at the *EIN2* locus, in which the depletion values are cut off. These values would allow an estimation of the signal to noise ratio and probably indicate that the ChIP signals for EEN oscillate around the background. There are potential reasons for EEN's weak enrichment over the background, e.g. if the INO80 complex is distributed with low affinity throughout the genome, associates very transiently with target regions or simply, more specific to EEN, this component, as part of a large protein complex may not be sufficiently cross-linked to DNA. The EEN ChIP-seq data require proper visualization and an indication of the significantly enriched regions within the graph for *EIN2*. The very weak binding of both EEN and REF6 at *EIN2* should also be discussed.

6) In Figure 4C and D, it would be useful to see these data as heatmaps in addition to average plots. Also, are these genes direct targets of INO80 and do they increase in expression in response to ethylene? Another potential interpretation is that INO80 is needed for the activation of these genes, and the loss of H2A.Z is really an effect of increased transcription.

[Editors' note: further revisions were requested prior to acceptance, as described below.]

Thank you for resubmitting your work entitled "Epigenetic silencing of a multifunctional plant stress regulator" for further consideration at *eLife*. Your revised article has been favorably evaluated by Christian Hardtke as the Senior Editor and Daniel Zilberman as the Reviewing Editor.

The manuscript has been improved but there is a remaining issue that needs to be addressed before acceptance:

Reviewers expressed serious concerns about the antibody used for H2A.Z ChIP experiments. This antibody was raised against a full-length budding yeast H2A.Z protein and its performance against Arabidopsis H2A.Z has not been evaluated. Several approaches to address this issue were discussed among the reviewers, with the least laborious being a thorough comparison to published H2A.Z ChIP-seq datasets. Hence the request in our decision letter: "At minimum, please include a comprehensive comparison of the H2A.Z ChIP-seq data to one or more of the publicly available Arabidopsis ChIP-seq datasets derived from well-validated H2A.Z antibodies."

The metaplot of genes comparing anti-H2A.Z and anti-GFP-H2A.Z profiles in Figure 2—figure supplement 1G and the genome browser snapshot in Figure 2—figure supplement 1H are not comparisons to validated, published data, nor do they constitute a comprehensive comparison. Therefore, please include a detailed analysis that would unambiguously demonstrate that your H2A.Z ChIP-seq data are comparable to published datasets. This should include correlation analyses, heatmaps, patterns of enrichment in unmethylated genes, methylated genes and transposons, snapshots of representative loci, and any other data that you feel would convince readers that the H2A.Z antibody used in this study performs appropriately.

---

## [Author Response]

Essential revisions:1) The interaction between H2A.Z and H3K27me3 was recently described in Arabidopsis, as the authors note: "H2A.Z and H3K27me3 are functionally linked and both repress transcription in Arabidopsis (Carter et al., 2018, Coleman-Derr and Zilberman, 2012)." However, the authors do not mention the work by Carter et al. further, including in the discussion of how H3K27me3 and H2A.Z might interact (Discussion). Carter et al. show that H2A.Z deposition in gene bodies is dependent on the H3K27me3 methyltransferase CLF, and H3K27me3 is promoted by the H2A.Z deposition factor PIE1. They propose a model for mutual reinforcement, where H2A.Z incorporation promotes H3K27me3 and H3K27me3 stabilizes H2A.Z by fostering nucleosome retention. Importantly, the work by Carter et al. indicates that the relationship between H3K27me3 and H2A.Z is quite general and not confined to a few genes that include EIN2. The authors should discuss their findings in light of this work, including in the model in Figure 5C. The integration of work by Carter et al. should also clarify the model, as the "blocking" connectors from EIN6 to SWR1 and from INO80 to PRC2 are not apparently supported by sufficient data. The authors may also wish to cite Dong et al., 2012, which noted an enrichment of H2A.Z associated with H3K27me3.

We thank the editor/reviewers for these comments and apologise for these oversights. In the revised manuscript, we now fully discussed the work by Carter et al. and we also include citation of the work by Dong et al., 2012. We agree that the functional linkage of H3K27me3 and H2A.Z is quite general and that their mutual reinforcement not only takes place at a few genes like *EIN2*. However, what makes *EIN2* relatively unique is the fact that two chromatin regulators (EIN6 and EEN) must both be functionally impaired, before the mutual reinforcement of H3K27me3 and H2A.Z deposition can be observed. Thus, the discussion was initially more focused on the missing regulatory steps that led to the increased mutual reinforcement of H3K27me3/H2AZ deposition at *EIN2*. We have now added a new paragraph that discusses the work of Carter et al. in more detail (see below) and have also added the mutual reinforcement of H3K27me3 and H2A.Z to our model legend. Moreover, we now include recent work from Gomez-Zambrano et al., 2019 into our Discussion. They report that monoubiquitination of H2A.Z is critical for H2A.Z-mediated gene repression.

We added this paragraph:

“H3K27me3 and H2A.Z strongly co-associate in *Arabidopsis* which is not surprising since the majority of *Arabidopsis* genes are marked with H2A.Z (Dong et al., 2012, Carter et al., 2018, Gómez-Zambrano et al., 2019). […] However, it is unclear why the increased mutually enforced deposition of H3K27me3 and H2A.Z can only be observed in the double mutant background. Although the functional loss of EIN6 (REF6) as the major *Arabidopsis* H3K27me3 demethylase has a profound impact on the H3K27me3 landscape in *Arabidopsis* (Lu et al., 2011), *EIN2* is only marginally affected.”

We have added this sentence to the legend of Figure 5C:

“When this EIN6 (REF6)/INO80-mediated regulation is functionally impaired, H3K27me3 and H2A.Z start to accumulate over the entire gene body of *EIN2* mutually reinforcing their depositions.”

2) The authors' results and those of Carter et al. indicate that K3K27me3 should be lost at EIN2 in ref6;een;pie1 triple mutants. The authors should evaluate this, as the loss of K3K27me3 at EIN2 is an important prediction of their model.

We agree with the reviewer that H3K27me3 ChIP-seq data at *EIN2* in *ref6-1 een-2 pie1-1* triple mutants would be helpful to prove our model. The triple mutant has a severe growth phenotype and produces only a few seeds per plant. Nevertheless we attempted to conduct H3K27me3 and H2A.Z ChIP-seq assays in etiolated Col-0 and *ref6-1 een-2 pie1-1* seedlings. We performed the ChIP assays with approximately 20 triple mutant seedlings which probably did not yield enough chromatin. Unfortunately, we were not successful in obtaining a significant ChIP-seq signal with *ref6-1 een-2 pie1-1* chromatin. We also used the Accel-NGS 2S Plus DNA Library Kit from Swift Biosciences which we usually use for more challenging cases like ChIPreChIPs or ChIPs with very low starting material. If requested, we can provide more detailed documentation of this experiment.

3) The H2A.Z antibody used in this study is a commercial preparation raised against the full-length budding yeast H2A.Z protein. The WT and pie1 ChIP-seq data strongly suggest that this antibody recognizes Arabidopsis H2A.Z, but given the importance of the data for the paper's conclusions, validation of the antibody and the ChIP-seq data is important. At minimum, please include a comprehensive comparison of the H2A.Z ChIP-seq data to one or more of the publicly available Arabidopsis ChIP-seq datasets derived from well-validated H2A.Z antibodies. A western blot using the antibody on WT and pie1 chromatin fractions would also be helpful.

In addition to the H2A.Z ChIP-seq data in Col-0 and *pie1* seedlings in Figure 2—figure supplement 2F, our manuscript also shows (Figure 2—figure supplement 2G) a comparison between the global H2A.Z occupancy in Col-0 *HTA11:HTA11-GFP* seedlings either determined with a commercial GFP antibody or the commercial H2A.Z antibody that we used for all our H2A.Z experiments. The Col-0 *HTA11:HTA11-GFP* line was obtained from Dr. Phil Wigge (Sainsbury Laboratory, Cambridge) and the *HTA11:HTA11-GFP* cassette has been proven to complement the *hta9hta11* double mutant. Both ChIP experiments were performed with equal amounts of the same *Col-0 HTA11:HTA11-GFP* chromatin to avoid any experimental variation. Figure 2—figure supplement 2G as well as the AnnoJ genome browser screenshot in Figure 2—figure supplement 2H clearly show that both ChIP experiments reveal similar genome wide occupancy patterns as well as single gene level. We have added a small paragraph into the main text for clarification. Specifically, we added this paragraph:

“To further validate our H2A.Z antibody, we performed ChIP-seq analyses with equal amounts of the same Col-0 *HTA11:HTA11-GFP* chromatin using either a commercial GFP or our commercial H2A.Z antibody. Both antibodies gave similar H2A.Z occupancy results (Figure 2—figure supplement 2G and H). “

4) The reviewers expressed concerns about the re-ChIP data-set. Visualization of these data (as provided in the authors' link) indicates that they are not very convincing and that the IgG control is not clean. Since the data are not extensively integrated into the major conclusions, we suggest removing them. Alternatively, please include an analysis that demonstrates the validity of these data.

As suggested, we have removed the re-ChIP dataset from the manuscript. We changed the figure legend title of Figure 2—figure supplement 2 to “*EIN2* displays an aberrant chromatin signature *ein6-1 een-1* mutants and is a direct target of EIN6.”

5) Related to point 4, the EEN-ChIP dataset did not show a convincing signal to noise ratio, despite the great number of peaks identified by SICER. In Figure 3A, the authors display a profile of enrichment over background at the EIN2 locus, in which the depletion values are cut off. These values would allow an estimation of the signal to noise ratio and probably indicate that the ChIP signals for EEN oscillate around the background. There are potential reasons for EEN's weak enrichment over the background, e.g. if the INO80 complex is distributed with low affinity throughout the genome, associates very transiently with target regions or simply, more specific to EEN, this component, as part of a large protein complex may not be sufficiently cross-linked to DNA. The EEN ChIP-seq data require proper visualization and an indication of the significantly enriched regions within the graph for EIN2. The very weak binding of both EEN and REF6 at EIN2 should also be discussed.

The binding of EIN6 and EEN to *EIN2* is indeed relatively weak compared to stronger binding sites such as EIN6 binding to *TCH4* (Figure 2—figure supplement 2J) and EEN binding to *AT4G01250* (Figure 3—figure supplement 1C). We think that this has a biological result rather than a technical issue since our EIN6, as well as our EEN ChIP-seq, have very convincing signal to noise ratios. SICER analyses clearly reflect the AnnoJ genome browser screenshots depicted in Figure 2—figure supplement 2J and Figure 3—figure supplement 1C. *AT4G01250* is among the top100 EEN targets and shows a 6.4 fold enrichment whereas *EIN2* is a relatively weak target (rank 3410) and shows a significant 1.66 fold enrichment. However, the specific enrichment of EEN is at the 3’ end of *EIN2* which overlaps with the increase of H2A.Z at the 3’ end in the double mutant background (Figure 2A, 3C; Figure 3—figure supplement 1C). We now indicated that information in Figure 3A and also modified the main text slightly:

“EEN also binds to *EIN2*, especially in the 3’UTR region (Figure 3A; Figure 3—figure supplement 1C).”

With regards to the weak binding of EIN6 and EEN to *EIN2*, we have added a new paragraph to the Discussion:

“This might also provide an explanation for the relatively weak binding of EIN6 and EEN to *EIN2*. Since we were able to capture strong EIN6 and EEN binding at numerous loci, technical reasons for weak binding such as insufficient crosslinking to the DNA are less likely. It is plausible that the number of cells with EIN6/EEN-binding at *EIN2* is rather small or that DNA residency times and chromatin remodeler compositions can vary between genes, thus potentially affecting the outcome of our ChIP-seq experiments.”

6) In Figure 4C and D, it would be useful to see these data as heatmaps in addition to average plots. Also, are these genes direct targets of INO80 and do they increase in expression in response to ethylene? Another potential interpretation is that INO80 is needed for the activation of these genes, and the loss of H2A.Z is really an effect of increased transcription.

The missing heatmaps were added as Figure 4—figure supplement 1D. Only 20% of the 1076 genes with a robust ET-induced H2A.Z eviction are direct EEN targets (227 genes of 1076 genes). One reason for the low overlap could be that our EEN ChIPs were not conducted under ET-inducing conditions as well as the potential very transient nature of INO8O/EEN binding at their target genes. We cannot rule out the reviewers hypothesis that INO80 is potentially important for the transcriptional activation of these genes and that the observed eviction of H2A.Z is just a consequence of increased transcription. We therefore have weakened our statement from “confirming” to “suggesting.” We also noticed some extra tick marks in Figure 4 which we have now removed.

Regarding the role of H2A.Z in the general ET response, our triple response phenotype data suggests that H2A.Z plays only a minor role. Neither *een, ino80, arp5* nor *pie1* mutants show a triple response phenotype. In addition, our transcriptome data for *een-1* under ET-inducing conditions data also suggests that H2A.Z eviction is not critical for a proper transcriptional response. However, we only profiled these plants at the 4 hours’ time point. It is possible that profiling expression kinetics would yield a clearer picture of the functional relationship between H2A.Z and ET-induced transcription. Another reason might be that the concentration of ethylene applied might be higher than occurring in nature thereby simply masking minor effects of INO80 complex mutations. Moreover, compensation effects of the mutants might also be a plausible scenario.

[Editors' note: further revisions were requested prior to acceptance, as described below.]The manuscript has been improved but there is a remaining issue that needs to be addressed before acceptance:Reviewers expressed serious concerns about the antibody used for H2A.Z ChIP experiments. This antibody was raised against a full-length budding yeast H2A.Z protein and its performance against Arabidopsis H2A.Z has not been evaluated. Several approaches to address this issue were discussed among the reviewers, with the least laborious being a thorough comparison to published H2A.Z ChIP-seq datasets. Hence the request in our decision letter: "At minimum, please include a comprehensive comparison of the H2A.Z ChIP-seq data to one or more of the publicly available Arabidopsis ChIP-seq datasets derived from well-validated H2A.Z antibodies."The metaplot of genes comparing anti-H2A.Z and anti-GFP-H2A.Z profiles in Figure 2—figure supplement 1G and the genome browser snapshot in Figure 2—figure supplement 1H are not comparisons to validated, published data, nor do they constitute a comprehensive comparison. Therefore, please include a detailed analysis that would unambiguously demonstrate that your H2A.Z ChIP-seq data are comparable to published datasets. This should include correlation analyses, heatmaps, patterns of enrichment in unmethylated genes, methylated genes and transposons, snapshots of representative loci, and any other data that you feel would convince readers that the H2A.Z antibody used in this study performs appropriately.

We have now added additional analyses to the manuscript to fully comply with the reviewer’s remarks. As suggested, we included three publicly available H2A.Z datasets (Carter et al., 2018, Wollmann et al., 2017, Coleman-Derr and Zilberman, 2012) into our analyses. Figure 2—figure supplement 1 was modified and contains now a correlation plot (Panel F), a heatmap (Panel G) and genome browser screenshots (Panel) that all show the suitability of the commercial H2A.Z antibody that we used in our study. Although our datasets are derived from 3 day-old etiolated seedlings, they are almost indistinguishable from the Carter et al., 2018 and the Wollmann et al., 2017 profiles. The Coleman-Derr and Zilberman, 2012 dataset is slightly different from ours which is probably caused by the difference in chromatin treatment (sonication vs. MNase).

We updated the respective paragraph in the main text and also included a new reference from our laboratory (Jupe et al., 2019) that uses the H2A.Z antibody from this study:

“Given the role of the INO80 complex in H2A.Z eviction (Alatwi and Downs, 2015, PapamichosChronakis et al., 2011), we employed ChIP-seq to survey the genome-wide distribution of H2A.Z using a commercial H2A.Z antibody (Jupe et al., 2019). […] In addition, a direct comparison of our antibody validation H2A.Z profiles (Col-0, *pie1-1*, Col-0 *HTA11:HTA11-GFP* (αH2A.Z)and Col-0 *HTA11:HTA11-GFP* (αGFP))with three publicly available H2A.Z ChIP-seq datasets (Carter et al., 2018, Wollmann et al., 2017, Coleman-Derr and Zilberman, 2012) further confirmed the suitability of our H2A.Z antibody (Figure 2—figure supplement 1F-H).”

We added an extra paragraph into the Materials and methods section:

“To generate aggregated profiles, heatmaps and correlation analyses of ChIP-seq data, Deeptools (Ramirez et al., 2014) was employed. […] multiBamSummary and plotCorrelation from Deeptools was used to generate the Spearman’s correlation plot. A total of 59814 2kb bins were found.”

We also updated the legend for Figure 2—figure supplement 1F-H:

“(F) Spearman’s correlation plot shows correlation of read coverages between the antibody validation H2A.Z datasets from this study (Col-0, *pie1-1*, Col-0 *HTA11:HTA11-GFP* (αH2A.Z)and Col-0 *HTA11:HTA11-GFP* (αGFP))andthree publicly available H2A.Z ChIP-seq datasets (Carter et al., 2018, Wollmann et al., 2017, Coleman-Derr and Zilberman, 2012). Clustering is determined by the degree of correlation. […] The shape difference of H2A.Z domains in the Coleman-Derr and Zilberman, 2012 dataset can be explained by the MNase treatment of the chromatin.”

We also noticed a typo in Figure 4—figure supplement 1F-H. This figure was updated and kilobase (kb) is now correctly abbreviated.